# Considerable yet contrasting regional imprint of circulation change on summer temperature trends across the Northern hemisphere mid-latitudes

Peter Pfleiderer[1], Anna Merrifield[2], István Dunkl[1], Homer Durand[3], Enora Cariou[4], Julien Cattiaux[4], Gustau Camps-Valls[3], and Sebastian Sippel[1]

[1]Institute for Meteorology, Leipzig University, Leipzig, Germany
[2]Institute for Atmospheric and Climate Sciences, ETH Zürich, Switzerland
[3]Image Processing Laboratory, Universitat de València, Spain
[4]Centre National de Recherches Météorologiques, Université de Toulouse, CNRS, Météo-France, Toulouse, France

**Correspondence:** Peter Pfleiderer (peter.pfleiderer@uni-leipzig.de)

**Abstract.** Rising summer temperatures and more frequent heat extremes are well-documented outcomes of anthropogenic climate change. However, the extent to which atmospheric circulation changes contribute to these trends remains contested. Regional differences across the northern mid-latitudes suggest that circulation plays a role, yet robustly quantifying its contribution over multiple decades is very challenging. We address this by systematically testing statistical and machine learning methods that decompose temperature signals into a thermodynamic and a dynamic contribution against climate model simulations. Specifically, we use unforced simulations with circulation nudged to match a forced simulation that includes anthropogenic emissions and land-use change. We apply decomposition methods to the forced simulations and compare their estimates of circulation-induced trends with those found in nudged circulation simulations. Our analysis reveals that most methods accurately identify the sign of circulation-induced changes in temperature, although they consistently underestimate their magnitude. Despite this limitation, the results demonstrate that circulation changes have made a substantial contribution to summer temperature trends across the northern mid-latitudes. In Europe, a hotspot region, we estimate that up to half of the observed summer warming between 1979 and 2023 can be attributed to circulation trends. Furthermore, circulation trends have contributed to warmer summer temperatures over Western North America, Central Siberia, Mongolia, Central China, and northeastern Canada. Yet, circulation changes have cooled summer temperatures over Eastern and Central North America, Eastern China, and Central Asia. Overall, our results, based on multiple methods, confirm a circumglobal mid-latitude pattern of considerable, yet contrasting, contributions of circulation changes to summer temperature trends.

## 1 Introduction

European summers have warmed considerably due to anthropogenic climate change, driving major ecological changes and societal impacts. This intensification of warm seasons is attributed directly to human activity (Seneviratne et al., 2021). The main reason for this warming is of a thermodynamic nature: warm seasons and heat extremes occurring in a warmer atmosphere are also warmer. However, also various other effects can contribute to regional trends in summer heat, including changes in

atmospheric circulation (Teng et al., 2022; Rousi et al., 2022; Vautard et al., 2023; Singh et al., 2023), or feedbacks due to land-atmosphere interactions (Seneviratne et al., 2006). As a result, regional trends in summer temperatures differ strongly across the world and even across the northern hemispheric mid-latitude land area.

In the mid-latitudes, large-scale circulation is a crucial driver for heat extremes (Rousi et al., 2022; Röthlisberger and Papritz, 2023) and there is great interest in understanding to what extent atmospheric circulation contributed to individual events (Cattiaux et al., 2010; Sippel et al., 2024; Zeder and Fischer, 2023), trends in heat extremes (Rousi et al., 2022; Singh et al., 2023) or seasonal temperatures (Teng et al., 2022). Forced changes in jet stream position and strength (Dong et al., 2022; Rousi et al., 2022; Woollings et al., 2023; Shaw and Miyawaki, 2024), changes in storm track intensity (Coumou et al., 2015;
Chemke and Coumou, 2024) and the resulting changes in weather pattern frequencies (Horton et al., 2015; Hanna et al., 2018; Fabiano et al., 2021) are likely to affect local climate conditions (Pfleiderer et al., 2019). Over the observational record, these forced changes, however, are small compared to internal climate variability (Eyring et al., 2021). Estimating the contribution of atmospheric circulation changes to local temperature trends and quantifying the extent to which these changes are due to forced or internal variability is crucial for our understanding of past summer temperature trends (Merrifield et al., 2017; Teng
et al., 2022; Vautard et al., 2023) and extreme events (Terray, 2021).

In some regions, the observed trends are falling outside the range of model-simulated expected trends (e.g., Western Europe, (Teng et al., 2022; Rousi et al., 2022; Vautard et al., 2023; Kornhuber et al., 2024)). Potentially, this indicates that model-simulated low-frequency variability in large-scale atmospheric circulation is too weak, or that a forced change in circulation is missing in the models, notwithstanding the broad uncertainty across models (Shepherd, 2014). The missing low-frequency
hypothesis is also challenging to assess because the available observations are relatively short. Understanding past circulation changes and their contribution to temperature trends may provide an opportunity to constrain future changes in summer temperature trends, acknowledging that future circulation changes remain a huge source of uncertainty (Topál and Ding, 2023; Fereday et al., 2018).

Identifying the contribution of atmospheric circulation to temperature trends is not straightforward (Deser et al., 2016).
Two main approaches are commonly used. The first applies statistical or machine learning methods to decompose observed or simulated temperature trends into thermodynamic and circulation-driven components, often referred to as dynamical adjustment (Deser et al., 2016; Smoliak et al., 2015; Sippel et al., 2019). The second uses nudged circulation simulations, in which the circulation is prescribed and the thermodynamic component is removed (Wehrli et al., 2018). Both options have their limitations: Nudged circulation simulations are limited by the representation of physical mechanisms in the climate model used
and might be subject to inconsistencies introduced by the nudging (discussed in section 4). On the other hand, most statistical decomposition methods are designed to capture the relationship between daily circulation patterns and daily temperatures. They do indeed capture day-to-day variability very well. Good skill is obtained on monthly or inter-annual time scales as well (Smoliak et al., 2015; Sippel et al., 2019; Cariou et al., 2025). Whether they can adequately capture a long-term trend is, however, more challenging to test, because processes determining long-term trends may be distinctly different from those
that determine short-term circulation variability, and much fewer verification samples are available. Moreover, benchmarks

for circulation-induced long-term trends have not been available to date, and to our knowledge, no systematic comparison of dynamical adjustment methods has been conducted.

In this study, we present a comprehensive assessment of circulation-induced summer temperature trends across the northern mid-latitudes (30N-60N) using both statistical decomposition methods and nudged circulation simulations. We focus on two key questions. First, can statistical-empirical methods reliably estimate circulation-driven long-term trends when tested against a climate model benchmark? To address this, we use a set of CESM2 nudged circulation simulations specifically designed to provide a reference for comparing circulation-induced trends with those derived from statistical methods. Second, we identify circulation-induced summer temperature trends across the northern hemispheric mid-latitudes in observations using four different statistical methods, as well as in CESM2 simulations that are nudged to the ERA5 circulation but driven without anthropogenic forcing.

## 2 Data & Methods

We use simulations of the fully coupled Community Earth System Climate Model, Version 2 (CESM2) (Danabasoglu et al., 2020), including simulations from the CESM2 large ensemble (Rodgers et al., 2021). Decomposition methods are first tested on nudged circulation CESM2 simulations. In section 3.2 we then apply decomposition methods to the European Centre for Medium-Range Weather Forecasts (ECMWF) Reanalysis v5 (ERA5) (Hersbach et al., 2020) for the period 1979-2023.

### 2.1 CESM2 nudged circulation simulations driven with CESM2 horizontal winds

To derive a benchmark for evaluating the decomposition methods, we use nudged circulation simulations conducted with the fully coupled CESM2 (Danabasoglu et al., 2020). First, three standard simulations following historical greenhouse gas emissions, aerosol emissions, and land use changes from 1850 to 2014, and anthropogenic forcings following the scenario SSP3-70 (O'Neill et al., 2016) from then onward (referred to as 'hist+SSP370') are created. These simulations follow the protocol of the CESM2 large ensemble (Rodgers et al., 2021). The initial conditions for these runs are taken from a long piControl (e.g., no anthropogenic greenhouse gas and aerosol emissions, as well as no land use change) simulation and they are separated by 100 years (namely year 1300, 1400 and 1500) to guarantee independent ocean states.

In a second step, for each of these simulations, a nudged circulation simulation is created for piControl forcing. Each of these simulations starts with the same initial conditions as its corresponding hist+ssp370 simulation. 6-hourly global meridional and horizontal winds are nudged globally and throughout the atmosphere (at all vertical levels) to their corresponding hist+ssp370 simulation. The nudging is achieved via a regular relaxation procedure described in the handbook of the Community Atmosphere Model version 6 (CAM6) (camdoc). The nudging procedure has been used in previous studies (Topál and Ding, 2023). These simulations are referred to as 'piControl-nudged', as they lack direct anthropogenic forcing. Still, through the nudging of atmospheric circulation, any potential anthropogenic forcing on horizontal wind fields is present alongside the internal circulation variability of the atmosphere from the hist+ssp370 simulation. Because land–atmosphere coupling influences near-surface circulation differently under varying climate forcings, it is recommended to apply nudging only at higher altitudes,

allowing surface winds to evolve freely when extreme events are to be studied (Wehrli et al., 2018; Merrifield et al., 2019). Although these effects may influence individual events, the spatial patterns of long-term circulation trends remain robust regardless of whether near-surface winds are nudged (Singh et al., 2025).

### 2.1.1 Conceptual interpretation of CESM2 nudged circulation simulations as a benchmark for dynamical adjustment

In a free-running, fully coupled climate simulation, local and regional temperatures are shaped by both thermodynamic forcing, here represented by global mean surface temperature (GMST) as a proxy for the large-scale thermodynamic background conditions, and atmospheric circulation variability, as well as their interactions (e.g., Deser et al. 2016, Fig. 1). Statistical dynamical adjustment methods seek to separate these influences, but their skill is challenging to evaluate within a coupled system. To address this, we use 'piControl-nudged' simulations as benchmarks in which circulation varies while thermodynamic forcing is fixed at pre-industrial levels. This design allows us to isolate and evaluate the circulation contribution independently of thermodynamic trends. Conceptually (Fig. 1), circulation variability is "inherited" from the parent simulation and is expected to dominate local and regional temperature responses. This inherited variability includes both internal fluctuations and possible forced circulation changes. Atmospheric circulation can affect GMST (arrow a in Fig. 1). Additionally, some internal variability unrelated to atmospheric circulation may remain, such as ocean-driven GMST fluctuations (arrow b in Fig. 1). Note that the decomposition into "thermodynamic" and "circulation-induced" changes is a simplification that overlooks important mechanisms for local temperatures. In Section 4, we discuss the implications of this simplification in more detail.

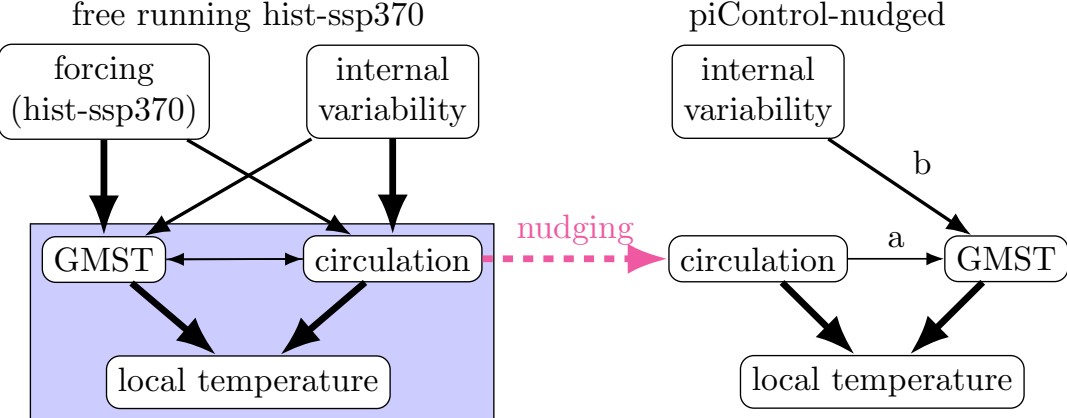

**Figure 1.** Conceptual illustration of causal relationships influencing local temperatures in a freely running climate simulation (left) and in a piControl-nudged simulation (right). The arrow width indicates the assumed importance of links. The blue rectangle highlights the processes studied here.

### 2.1.2 Illustration of CESM2 nudged circulation simulations

As expected, day-to-day variability in the nudged circulation run is closely related to its freely running counterpart from which the wind fields originated. As shown in Figure A1, in the early period (left column of fig. A1), geopotential height at 500

hPa and surface air temperature are nearly identical in the hist+ssp370 and the piControl-nudged run. In a warmer climate, day-to-day variability remains highly correlated, but geopotential height and surface air temperatures are uniformly shifted to higher values.

Interpreting the GMST signal in the piControl-nudged runs is not straightforward. Even without external forcing, small GMST trends emerge over 40-year periods (Fig. 2b,c). These trends likely reflect the combined influence of atmospheric circulation and internal ocean variability (see Fig. 1).

A comparison of 1979-2023 GMST trends in the piControl-nudged simulations with their corresponding freely running hist-ssp370 simulations indicates that both thermodynamic and dynamic contributions are relevant (Figure 2b). Run 1 is
the simulation with the highest GMST trend in the freely running configuration (0.25 K/decade) and near zero trend in the piControl-nudged configuration. Runs 2 and 3 have weaker GMST trends than Run 1 in the freely running hist-ssp370 scenario, and both show a cooling trend in the piControl-nudged scenario. The differences between freely running hist-ssp370 simulations and their corresponding piControl-nudged simulations are not constant, indicating that there is indeed an influence of low frequency internal variability that is not controlled by atmospheric circulation. Therefore, we do not expect
the circulation component estimated from the freely running forced simulation to match the trend found in the piControl-nudged simulation exactly. In some decomposition methods (ridge regression and DEA), we can account for the effect of these GMST trends in the piControl-nudged simulations. In general, we assume that the impact of GMST on local temperatures is relatively homogeneous around the mid-latitudes of the northern hemisphere. Thus we expect that the spatial pattern relative to the mid-latitudinal mean is well captured.

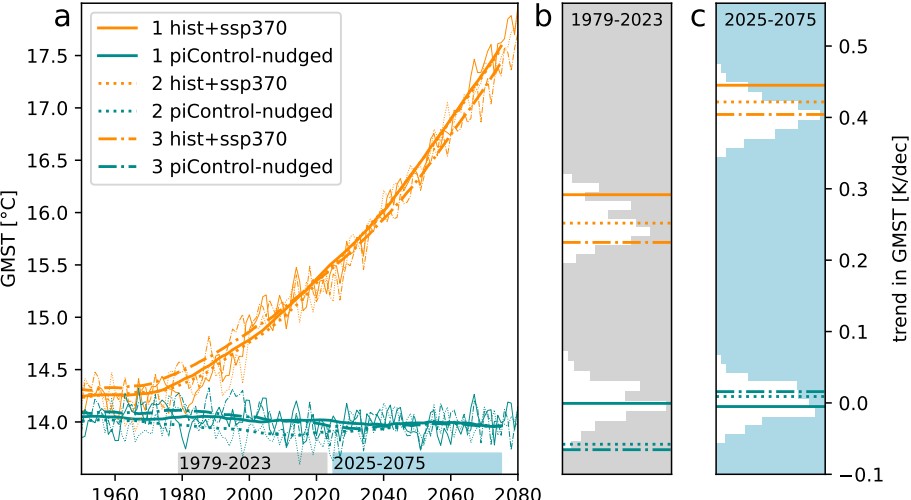

**Figure 2.** a: Global mean surface temperature in hist+ssp370 runs (orange) and in piControl-nudged runs (green). Thick lines are smoothed with a 50-year rolling mean. b: GMT trend over the period 1979-2023 (gray bar in a). c: GMT trend over the period 2025-2075 (blue bar in a). The cooler histogram displays 500 trends of the same length from piControl, while the warmer histogram displays 100 trends from the CESM2 large ensemble.

## 2.2 CESM2 nudged circulation simulations driven with ERA5 winds

To evaluate the use of the decomposition methods on observed circulation patterns, we created an additional benchmark simulation by nudging CESM2 to the horizontal wind fields from the reanalysis data ERA5 (Hersbach et al., 2020). This method relies on the same relaxation procedure and anthropogenic greenhouse gas and aerosol forcing as described before, with some modifications to the setup. Here, not only the piControl, but also the 'hist+SSP370' simulations are nudged to ERA5 horizontal wind fields between 1940 and 2024. There are two further differences in the simulation setup as opposed to the already described nudged circulation simulations. First, to account for forcing-induced variability at the boundary later, the atmosphere is only nudged above 700 hPa, similar to (Wehrli et al., 2018). Second, the model is run in an atmosphere only configuration (AMIP) with prescribed ocean conditions from the Met Office Hadley Centre's sea ice and sea surface temperature dataset (HadISST; Rayner et al. 2003). For the piControl simulation, the forced component was removed from HadISST using low-frequency pattern filtering (Wills et al., 2020). Sea ice concentration (SIC) was then estimated with a random forest model trained on the SST–SIC relationship in HadISST. To capture seasonal hysteresis, separate models were trained for the freezing season (Sept–Jan) and the melting season (Feb–Aug), and applied to piControl SSTs to generate corresponding SIC values. This approach reproduces daily temperature anomalies in the northern hemisphere well. However, nudging to observed winds and SSTs also introduces a temperature bias by displacing the model from its own climatology.

## 2.3 Dynamical adjustment methods

We test four methods designed to disentangle the dynamic effect from the thermodynamic effect: (1) ridge regression, (2) constructed circulation analogues, (3) direct effect analysis (DEA), and a neural network UNET (4).

Note that we apply the methods exactly as they were designed and used in other publications, and therefore different proxies for atmospheric circulation are used by different methods. We do not expect that the choice of variable to represent atmospheric circulation significantly affects the results. In Figure G1, we show a sensitivity analysis for the ridge regression. In Section 4, we discuss differences between the methods and how they might affect the decomposition in more detail.

### 2.3.1 Ridge regression

Ridge regression is a regularized linear method that can deal with high-dimensional predictors (in our case, many correlated spatial locations with streamfunction values, $\Phi_j$). For each grid cell, we train a ridge regression model to predict daily mean temperatures in summer (June-July-August, hereafter referred to as JJA), $Y$, using GMST (yearly averaged) and the streamfunction $\Phi$ at 500 hPa in all grid cells $G$ within a $40 \times 40$-degree rectangular area centered on that grid cell:

$$Y = \gamma_0 + \gamma_1 GMST + \sum_{j=2}^{G+1} \gamma_j \Phi_j + \epsilon, \tag{1}$$

where the streamfunction ($\Phi$) is related to horizontal eastward ($u$) and northward ($v$) wind speeds as $u = \frac{\partial \Phi}{\partial y}$ and $v = \frac{\partial \Phi}{\partial x}$.

Ridge regression mitigates overfitting by introducing a penalty for model complexity, achieved through the shrinkage of regression coefficients related to the streamfunction covariates. Shrinkage is determined by the sum of squared regression

coefficients (known as L2 regularization) and a regularization parameter $\lambda$, which controls the degree of shrinkage. The shrinkage term is added to the residual sum of squares ($RSS$) for the minimization:

$$\hat{\gamma} = \underset{\gamma}{\mathrm{argmin}}\{RSS + \lambda \sum_{j=2}^{G+1} \gamma_j^2\}. \tag{2}$$

As a result, ridge regression solves a joint minimization problem, producing small but nonzero regression coefficients that are relatively evenly distributed among correlated predictors. The regularization parameter $\lambda$ dictates the extent of shrinkage and is selected via cross-validation or knowledge about the noise variance. Notably, the intercept of the linear model as well as the GMST covariate remain excluded from the shrinkage.

### 2.3.2 Constructed atmospheric circulation analogues technique

The atmospheric circulation analogue technique, introduced in Deser et al. (2016), is a linear dynamical adjustment method. It is designed to provide empirically derived estimates of climate trends induced by "dynamics" or atmospheric circulation patterns. It achieves this through the re-construction of monthly mean climate fields by linear regression with coefficients derived from a field representative of atmospheric circulation (here, sea-level pressure). The method has been used for a variety of applications, including trend assessments, variability analysis, performance weighting, and extreme event attribution (Deser et al., 2016; Lehner et al., 2017; Merrifield et al., 2017; Guo et al., 2019; Terray, 2021).

The method is based on the construction of a target monthly mean sea level pressure (SLP) field (e.g., January 1980) using analogues. In the January 1980 example, analogues are SLP fields from other Januaries between 1850 and 2014 that resemble the target SLP pattern. The method is applied to every month in the record and proceeds as follows. First, the Euclidean distance between all SLP fields from the period of 1850-2014 and the target SLP field is computed. Euclidean distance is calculated at each grid point and averaged over the Northern hemisphere domain (20–90 °N, 0–360 °E). The $N_a = 80$ closest SLP fields to the target are considered analogues. From the 80 analogue choices, the target month is reconstructed using randomly selected subsets of $N_s = 50$ analogues. The process of choosing 50 out of 80 analogues and reconstructing the target SLP is repeated $N_r = 100$ times to obtain an average best estimate result. $N_a$, $N_s$, and $N_r$ values are consistent with those used for the method's application to observations in Deser et al. (2016).

The target SLP field $\mathbf{X}_h$ is reconstructed through multivariate linear regression. The weight assigned to each SLP analogue, $\beta$, is computed through a singular value decomposition of a column vector matrix $\mathbf{X}_c$ containing the 50 selected analogues and can also be estimated using a Moore-Penrose pseudoinverse:

$$\beta = [(\mathbf{X}_c^T \mathbf{X}_c)^{-1} \mathbf{X}_c^T] \mathbf{X}_h \tag{3}$$

where $\beta$ weights are applied to the corresponding monthly mean temperature fields, i.e., those from the same month as the SLP analogue. The weighted linear combination of these fields defines the dynamic component of temperature for the target month. Before weighting, a quadratic trend is removed from the full temperature record at each grid point to approximate the anthropogenic warming signal (Deser et al., 2016; Lehner et al., 2017). This is done to approximate the unforced relationship

between SLP and temperature one would expect to find in a preindustrial control simulation. The dynamic component of temperature is also computed $N_r = 100$ times for each month. Results are averaged, and once every month, the record is dynamically adjusted as described; we obtain a dynamic monthly mean temperature timeseries.

Note that the Moore–Penrose pseudoinverse implicitly deploys a "hard" regularization (i.e., kills directions with singular value exactly zero). In contrast, the previous ridge (Tikhonov) regularization imposes an explicit "soft" regularization (i.e., damps unstable directions even if singular values are just small, not zero).

    We use the term "analogue" to refer to a month with an SLP field close to the SLP target. The Euclidean distance selection metric does not require an analogue to match the target month spatially over the whole domain. This step is necessary because,

with fewer than 200 available analogues, it is improbable to find a perfect match for any target month. Van Den Dool (1994) estimated that it would take on the order of $10^{30}$ years to find two Northern Hemisphere circulation patterns similar within observational uncertainty. As a result, the method relies on imperfect analogues, which can introduce spurious features or bias the amplitude of the estimated dynamic temperature trends.

    In this study, analogues are selected from the free-running hist+SSP370 simulations to dynamically adjust (1) each hist+SSP370

simulation and (2) ERA5. Each hist+SSP370 simulation is dynamically adjusted using the "leave-one-out" approach (Deser et al., 2016; Lehner et al., 2017). In the leave-one-out approach, for each month, e.g., June 1900, analogues are selected from all other Junes in the simulation's 1850-2014 period except 1900. The leave-one-out approach is used for the comparison between the hist+SSP370 and piControl-nudged simulations. In the second approach, analogues are selected from the entire 1850-2014 period of each of the hist+SSP370 simulations and used to dynamically adjust ERA5. The resulting three dynamical

components of ERA5 are shown in Figure F1 and are averaged to produce the circulation-induced trend estimates in Figure 4.

### 2.3.3   DEA

We employ Direct Effect Analysis (DEA), a recently developed causal representation learning method, to separate the dynamical influence of atmospheric circulation (represented by Z500 EOFs) from the thermodynamical influence of GMST on temperature. The approach aims at disentangling an outcome variable $Y$ into a direct effect component, $Y_{\text{dir}}$, which represents the part of $Y$

directly caused by some causal factor $Z$, and an orthogonal component, $Y_{\text{orth}}$, which corresponds to the part of $Y$ unaffected by $Z$. Both $Z$ and $Y$ may be influenced by other variables $X$, which act as confounders, and it is thus necessary to control for these covariates to get a correct estimate of the direct effect of $Z$ on $Y$. This learned representation of $Y$ can be seen as the result of encouraging conditional independence between $Z$ and $Y$ while controlling for $X$ (Durand et al., 2025).

    In this context, the outcome $Y$ represents the temperature field—$Y_{\text{orth}}$ being its dynamical component—which is a random

vector with one dimension per grid cell. The predictor $Z$ is the monthly mean GMST, used as a proxy for thermodynamic temperature changes. As covariates $X$, we include the leading Empirical Orthogonal Functions (EOFs) of atmospheric circulation (Z500), denoted $\{p_j\}_{j=1}^{J}$ where $J$ is selected through 5-fold cross-validation to maximize the $R2$ score.

Similar to the ridge regression approach described above, we assume the following linear model:

$$Y = b_0 + \mathbf{b}_1 \text{GMST} + \sum_{j=2}^{J+1} \mathbf{b}_j p_j + \epsilon. \tag{4}$$

and get the optimal regression parameter matrix using a least squares algorithm. We obtain a matrix $\mathbf{B}$ whose columns $\mathbf{b}_i$ encode how GMST and each EOF influence temperature across grid cells. We emphasize that this is a multivariate regression problem, where $Y$ is a random vector—not a single variable—representing temperature values across multiple grid cells. Each dimension of $Y$ corresponds to one spatial location.

To isolate the dynamical component $Y_{\text{orth}}$ of $Y$, we remove the part aligned with the GMST-related direction, as captured by $\mathbf{b}_1$. This is achieved using the linear transformation $Y_{\text{orth}} = \mathbf{P}^\top Y$, where $\mathbf{P} = \mathbf{I} - \frac{\mathbf{b}_1 \mathbf{b}_1^\top}{\|\mathbf{b}_1\|^2}$. This transformation ensures that $Y_{\text{orth}}$ remains unaffected by any interventions on GMST, and thus represents the dynamical component of $Y$.

### 2.3.4 UNET

The final method used in this paper is a convolutional neural network, a UNET structure, recently proposed by Cariou et al. (2025) to link temperature variations to atmospheric circulation. The UNET architecture was initially introduced by Ronneberger et al. (2015) for biomedical image segmentation. It consists of two main components: an encoder and a decoder. The encoder extracts global features from the input (in this case, circulation maps) by progressively reducing spatial resolution while increasing feature depth through convolution and max-pooling layers. The decoder then reconstructs the image using transposed convolutions. Symmetry between the two parts, combined with skip connections, allows the network to preserve and effectively reuse encoded information.

We use this architecture (see E1) to estimate the part of daily temperature variations ($T'$, the output) which can be explained by the large-scale circulation, described by the sea level pressure (SLP, the input). Thus, we can write the UNET model as

$$T' = F(SLP). \tag{5}$$

We follow the methodology described in Cariou et al. (2025). Still, we extend the analysis to a larger spatial domain and train the UNET on daily data from 1850 to 2100 from 8 CESM2 transient simulations (80% of the data are randomly selected for training, and the remaining 20% are used for validation). These simulations belong to the CESM2 ensemble, but differ from the three members used to build the nudging experiment (referred to as run 1, run 2 and run 3) in order to prevent overfitting by the UNET. Since the UNET is trained on transient runs (historical and SSP), we must consider climate change in the relationship 5. The SLP is not detrended, assuming that in the CESM2 model, the forced responses in the SLP is small compared to the daily variability. This assumption is supported by Figures 2 and 3, which show that the three piControl-nudged simulations do not exhibit significant common trends. However, the forced response is substantial in terms of temperature. Thus, the detrending is made following Rigal et al. (2019): temperature anomalies (T') are obtained by removing an estimate of the daily non-stationary normal containing both the mean seasonal cycle, which is not circulation-explained, and the climate change signal. The trained model is then tested on three CESM2 piC-nudged runs, with SLP maps standardized using the same values as in the training process.

For ERA5, we use the UNET that was previously trained on CESM2 and retrain it on ERA5 data from 1940 to 1978. This process is known as a fine-tuning method. This two-step approach is motivated by the limited amount of ERA5 data available for training. Pre-training on CESM2 simulations allows the network to learn robust large-scale circulation-temperature relationships from a wide range of situations. The subsequent fine-tuning on ERA5 then adjusts these learned features to the characteristics of ERA5 data studied. SLP maps are standardized with mean and standard deviation calculated on this training period, and the non-stationary normal is computed thanks to an estimate of the forced response obtained with Qasmi and Ribes (2022) method. The inference is then done over the 1979-2023 period.

## 3 Results

We evaluate each statistical method's estimate of circulation-induced mid-latitude JJA temperature trends. Estimated trends $\hat{y}$ are derived from CESM2 free-running hist+ssp370 simulations and compared against trends in CESM2 piControl-nudged simulations $y$, which serve as the benchmark. Method performance is assessed using four skill metrics:

1. The percentage of grid-cells for which the trend sign is correctly identified. This metric provides a general sense of whether the method can correctly capture the sign of the trend, which may be sufficient in specific contexts—for example, in climate change detection.

2. Pearson correlation (pattern correlation across the mid-latitudes). Pearson correlation reflects how well the method captures the spatial pattern of the trend. Some methods may systematically over- or underestimate the magnitude of trends, yet still accurately reproduce their spatial distribution.

3. The coefficient of determination ($R^2 = 1 - \sum (y - \hat{y})^2 / \sum (y - \bar{y})^2$). $R^2$ is a widely used metric for spatial comparisons, as it accounts for the variance at each location and indicates how much of the observed variability is explained by the prediction. Yet, it is, in contrast to Pearson correlation, sensitive to any bias in the estimated average (Kvålseth, 1985); and hence, a statistical method may show a good spatial Pearson's correlation in its estimates but a poor R-squared score.

4. The regression slope between predicted and benchmark trend estimates. The regression slope indicates whether the method tends to overestimate or underestimate the magnitude of trends.

Note that all metrics are directly applied to the gridded data without any area weighting.

### 3.1 Evaluation of circulation-induced trends in the historical period (1979-2023) in CESM2 nudged-circulation simulations

Over the period 1979-2023, JJA temperature trends in the piControl-nudged simulations range from -0.35 to 0.35 K per decade (Figure 3 a,d,g). These trends are organized in large regional clusters of alternating signs. Furthermore, the trend patterns differ considerably between the three piControl-nudged runs, indicating that in CESM2, circulation-induced trends are dominated by internal variability and that in CESM2, forced circulation changes are minor. Overall, JJA temperature trends are slightly

stronger in run 1, which is likely due to the positive GMT trends in the piControl-nudged runs during this period (see Figure 2).

Note that most of these trends are not statistically significant (see Figure B1). Since these trends mostly reflect internal climate variability, it is expected that, from a statistical point of view, the circulation-induced temperature changes at one location are not differentiable from noise. The spatially consistent trend patterns indicate that, although lacking statistical

significance, these trends contain valuable information and are worth evaluating.

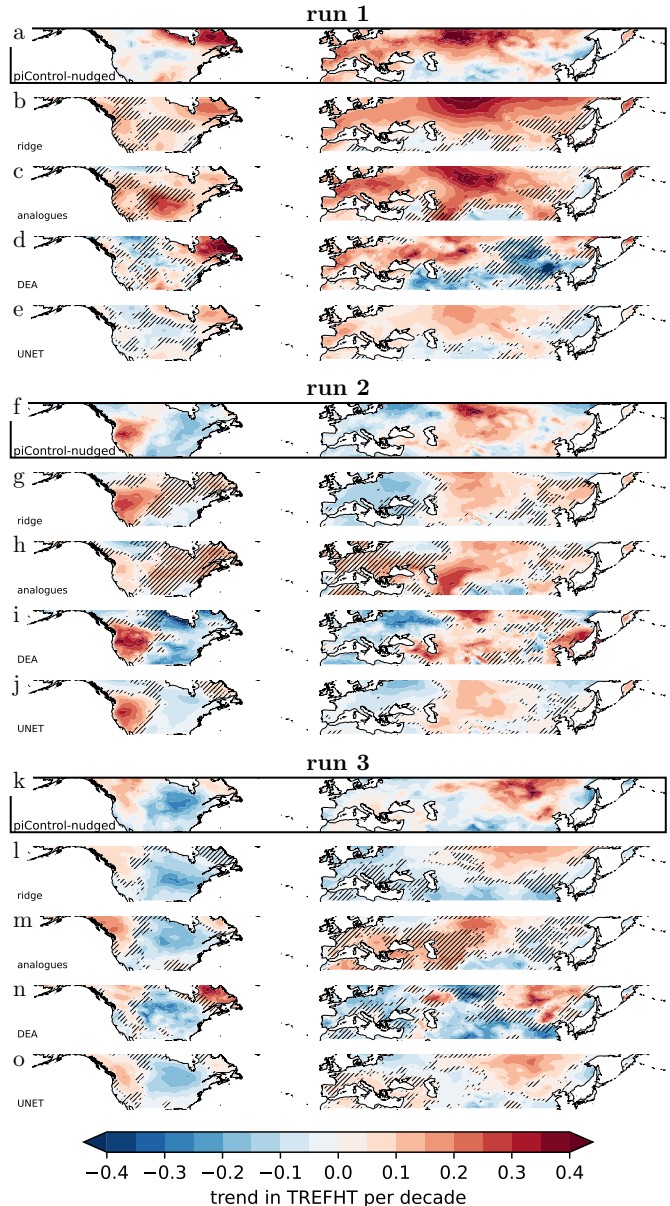

**Figure 3.** Trend in JJA temperatures over the period 1979-2023 in piControl-nudged (a,f,k) and predicted trends from different decomposition methods. For runs 1 (a-e), 2 (f-j), and 3 (k-o). Estimates from the ridge regression (b, g, l), the analogues (c, h, m), DEA (d, i, n), and UNET (e, j, o). Areas where the predicted trend differs in sign from the piControl-nudged run are highlighted by black hatching.

**Table 1.** Evaluation metrics comparing trends in piControl-nudged simulations to estimates of circulation induced trends from statistical decomposition methods for land grid-cells between 30N-60N and the period 1979-2023. First block: percentage of grid-cells with correctly predicted signs. Second block: Pearson correlation coefficient. Third block: Coefficient of determination. Fourth block: regression slope (as shown in Figure C1). See table D1 for the same evaluation over the period 2025-2075.

| run | ridge | analogues | DEA | UNET |
|---|---|---|---|---|
| **correct sign** | | | | |
| **all runs** | **75%** | **65%** | **76%** | **84%** |
| run 1 | 75% | 77% | 73% | 86% |
| run 2 | 74% | 56% | 82% | 84% |
| run 3 | 77% | 63% | 74% | 83% |
| **Pearson correlation (r)** | | | | |
| **all runs** | **0.75** | **0.52** | **0.64** | **0.86** |
| run 1 | 0.79 | 0.57 | 0.61 | 0.91 |
| run 2 | 0.67 | 0.36 | 0.74 | 0.83 |
| run 3 | 0.75 | 0.41 | 0.58 | 0.89 |
| **coefficient of determination (R2)** | | | | |
| **all runs** | **0.53** | **0.07** | **0.08** | **0.66** |
| run 1 | 0.54 | 0.18 | -0.19 | 0.50 |
| run 2 | 0.39 | -0.28 | 0.22 | 0.67 |
| run 3 | 0.48 | -0.07 | -0.08 | 0.73 |
| **regression slope** | | | | |
| **all runs** | **0.66** | **0.43** | **0.74** | **0.51** |
| run 1 | 0.59 | 0.47 | 0.71 | 0.47 |
| run 2 | 0.54 | 0.26 | 0.97 | 0.57 |
| run 3 | 0.55 | 0.36 | 0.68 | 0.58 |

Using the ridge regression trained on a forced simulation to predict the trends based on the streamfunction of the forced simulation and the GMT of the piControl-nudged run, we get a similar trend pattern as in the piControl-nudged run (Figure 3 b,e,h). Over the mid-latitudinal land area, half of the variability in local temperature trends in the piControl-nudged run is explained by the ridge regression model (compare R2 score in table 1). For three-quarters of the grid-cells, the sign of the

predicted trend is correct, and grid-cells for which the sign of the trend is not indicated correctly are mostly grid-cells with small trends in the piControl-nudged simulation (and the prediction).

The analogue method reveals a positive correlation between predicted and simulated mid-latitude land trends, with a similar percentage of correctly identified signs in Run 1 and slightly lower skill in Runs 2 and 3. Importantly, the analogue method only captures the circulation-driven component of the trend and does not account for GMST contributions in the piControl-nudged simulations. Consequently, performance is lower in runs 2 and 3, where substantial negative GMST trends were simulated. It is important to note, however, that an offset largely influences the relatively poor R2 score in the mean circulation trend (overestimated warming), while the spatial pattern itself shows a rather good resemblance and Pearson correlation compared to the benchmark simulation (Figure 3).

The DEA method performs well in estimating the sign of circulation-induced trends with 76% accuracy (see Table 1). Despite the relatively strong correlation between the trend maps (r = 0.64), the coefficient of determination is close to zero. Estimates of circulation-induced trends from DEA cover the full range of simulated piControl-nudged trends, including very high and very low trends. This is reflected by a relatively high regression slope between predicted and simulated trends (Figure C1 and last block in table 1).

The UNET is performing the best of all tested methods here. With UNET, 84% of trend signs are predicted correctly; it has the highest Pearson correlation coefficient (0.86) and the highest coefficient of determination (0.66). In comparison to the DEA and ridge regression, UNET tends to predict weaker circulation-induced trends and rarely exceeds magnitudes of 0.2 K per decade. As shown in Figure C1, this leads to a systematic underestimation of the trend magnitude compared to the piControl-nudged simulation.

The evaluation over a different period (2025-2075) yields similar skill metrics and confirms the above-discussed results (see Figure D1, D2 and Table D1).

Overall, UNET is the most accurate method for explaining the variance in mid-latitude boreal summer temperature trends. The ridge regression and UNET tend to decompose temperature trends into a regionally smoothed pattern of circulation-induced temperature trends, with a lower likelihood of predicting a strong trend of the wrong sign. DEA and the analogue method project strong trends of the wrong sign in some regions. DEA appears to be more helpful in estimating the potential magnitude of circulation-induced trends, whereas UNET is relatively conservative, estimating trends that are generally too weak. The analogue method also shows skill in predicting the sign of trend, but appears typically less trustworthy then other decomposition methods.

### 3.2 Identification of the circulation-induced boreal summer temperature trend (1979-2023) in reanalysis

Across both the ERA5 reanalysis and the CESM2 simulations nudged to ERA5 winds, all decomposition methods reveal
similar circulation-induced trend patterns for 1979–2023. Over Eurasia, a wave-like structure emerges: strong warming over
Central and Eastern Europe (around 30°E), cooling over Kazakhstan and western Siberia (60°–90°E), and warming again over
Mongolia, eastern Siberia, and Central China (90°–120°E), extending toward the Kamchatka Peninsula (Fig. 4). Across North
America, we observe a dipole pattern with positive trends in the western part and negative trends in the central and eastern parts,
with positive trends again in the outermost northeastern parts. All decomposition methods as well as the CESM2 simulations
nudged to ERA5 winds (Figure 4j) agree on this broad trend pattern with only little regional deviations. The trend pattern
identified with the statistical method is in good agreement with the pattern found in Teng et al. (2022).

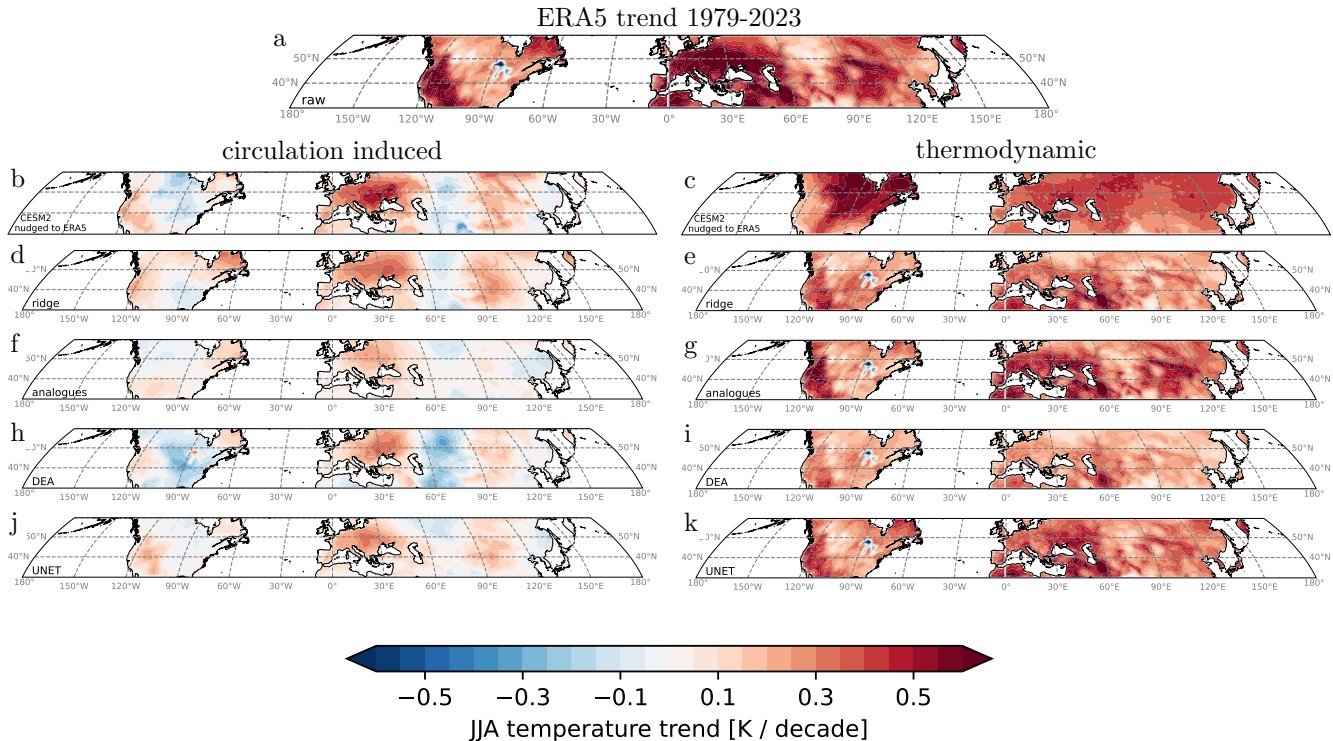

**Figure 4.** JJA mean temperature trends in ERA5 over the period 1979-2023 (a) decomposed in the circulation-induced (left) and
thermodynamic (right) contribution for the ridge regression (d,e), DEA (f,g), analogues (h,i), UNET (j,k) and estimates from CESM2
simulation with horizontal winds nudged to ERA5 winds (b,c).

Based on our evaluation of decomposition methods against dedicated nudged simulations in the CESM2 setup, we would
suggest giving more weight to the results from UNET regarding the sign of circulation-induced trends. For example, this
would imply that the positive circulation induced trends over northern North America in the ridge regression are probably
wrong (compare low skill of ridge regression in this region, Figure 3).

Moreover, there remains ambiguity on the magnitude of the above-described trend pattern. The ridge regression, DEA, and UNET suggest a circulation-induced trend of up to 0.3 K/dec over eastern Europe. At the same time, the piControl simulations from CESM2, which were nudged to ERA5 winds, show stronger circulation-induced trends of up to 0.6 K/dec. In other regions, the nudged simulations and DEA exhibit stronger trends, followed by the ridge regression and analogues, while UNET suggests somewhat weaker trends. From the evaluation of the decomposition methods, we know that all methods have indeed a tendency to underestimate the magnitude of circulation-induced trends somewhat, suggesting that the circulation-induced trend over eastern Europe could be around 0.5 K/dec as indicated by the nudged simulations.

In summary, our study confirms the highly variable mid-latitude boreal summer trend pattern found in Singh et al. (2023); Teng et al. (2022); Vautard et al. (2023) with five independent methods (four statistical methods and a nudged circulation simulation driven by ERA5 horizontal wind fields). The trend pattern highlights several regional warming hotspots where circulation has made a major positive contribution (Teng et al., 2022). While total boreal summer temperature trends are positive across the NH mid-latitudes, circulation has driven cooling in large areas—most notably Central and Eastern North America, Central Eurasia, and, to a lesser extent, coastal eastern China. In these regions, the moderately positive total trends reflect a compensation between circulation-induced cooling and thermodynamic warming.

## 4   Discussion

Many studies have sought to isolate circulation-induced components in climate time series, often referred to as dynamical adjustment (Smoliak et al., 2015; Deser et al., 2016; Guo et al., 2019; Cariou et al., 2025; Singh et al., 2023; Saffioti et al., 2017; Lehner et al., 2017). The core assumption is that circulation variability, primarily driven by internal processes, dominates temperature variability in many regions, while thermodynamic contributions can be derived as the residual (e.g. Deser et al., 2016). This separation of dynamic and thermodynamic components provides a powerful framework for climate attribution (e.g. Shepherd, 2014).

However, while different statistical methods for obtaining circulation-induced components in climate time series are routinely evaluated on short time scales, the estimation of circulation-induced decadal trends has remained a challenge for the climate community and will likely continue to do so. This is because of five main reasons.

First, statistical methods have been found to perform very well on short time scales of day-to-day, month-to-month, or inter-annual variability (Cariou et al., 2025; Smoliak et al., 2015; Sippel et al., 2019). Yet, the difference in the performance on long (that is, trend) time scales versus short time scales has not been quantified so far. Nevertheless, dynamical adjustment has been widely applied on the time scales of trends. A reduced performance on long time scales is expected, and indeed found in this study, because shorter time scales are dominated to the largest extent by circulation-induced variability, whereas on longer time scales other processes are becoming more dominant, such as land-atmosphere interactions (e.g. Merrifield et al., 2017) or long-term warming, both of which may not be straightforward to account for.

Second, and partly related to the previous point, designing a method comparison for the identification of circulation-induced time series is challenging. This is because it is not immediately apparent what the components are that the signal

is decomposed into, and which relevant mechanisms can be attributed to these components. In this study, we decompose a trend in local temperatures into a "circulation-induced" component and a "thermodynamic" component without specifying to which of these components changes in other important factors, such as soil moisture or aerosol concentrations, are attributed (see Figure 5 for the example of land-atmosphere feedbacks). Therefore, the different methods also estimate different trends for the "thermodynamic" component as shown in Figure 4. The different statistical methods evaluated here were initially developed for similar but slightly different research questions: The analogue method, for instance, was designed to separate the "thermodynamic signal" from "circulation-induced variability." Yet, it has been shown that summer land-atmosphere interactions remain largely in the residual, thermodynamic component due to the method's setup (Merrifield et al., 2017). On the other hand, machine learning methods such as the UNETs may implicitly identify land-atmosphere interactions as part of circulation variability, if circulation carries an imprint of land-atmosphere variability. In table 2 we summarize our thoughts on the treatment of land-atmosphere feedbacks in the different approaches.

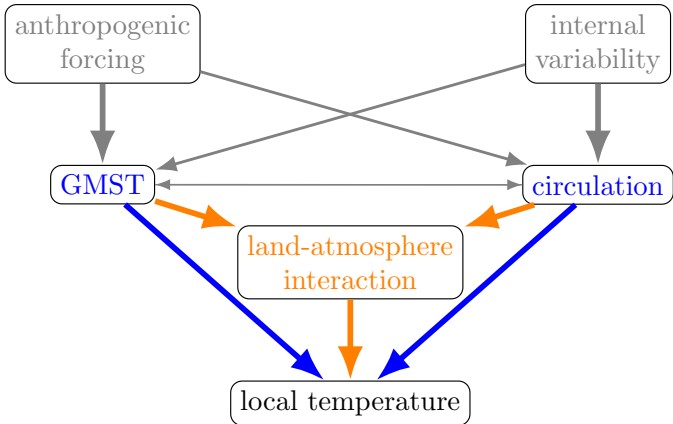

**Figure 5.** Conceptual illustration of causal relationships influencing local temperatures in a forced climate as in Figure 1 but with an additional driver of local temperature. In this study we aim at decomposing the influence on local temperature into thermodynamic (GMST) and circulation induced contributions (in blue). Land-atmosphere interactions is a driver we do not explicitly model (in orange).

Third, designing a benchmark for the circulation-induced component of climate time series, such as summer temperatures, is a challenging task. In this study, we use a piControl nudged-circulation approach, where a climate model is nudged to the horizontal winds of a forced transient simulation. This setup provides circulation-induced changes within an otherwise unforced climate simulation. This might lead to inconsistencies between wind fields and their drivers. For example, Arctic amplification and the resulting reduced equator to pole temperature gradient in the hist+ssp370 simulations might lead to weaker westerlies in the northern hemispheric mid-latitudes that would be imposed on the pre-industrial climate. There may also be factors of residual climate variability (such as ocean variability) or feedbacks between circulation and other factors, such as land-atmosphere coupling, that could still affect thermodynamical processes on climate over land (see table 2). Additionally, summer temperatures in the nudged circulation simulations might be affected by nudging in other seasons. For example, circulation changes can influence soil moisture in late spring which would then have an impact on summer temperatures. Statistical

decomposition methods do not use this information. Consequently, we have to admit that the nudged simulations are not a perfect benchmark. Further analysis is required to understand how these limitations affect our estimates of circulation-induced trends and whether a better-suited benchmark test could be designed.

**Table 2.** Expectations on how land-atmosphere interactions might influence the decomposition into "circulation-induced" and "thermodynamic" contributions.

| method | what is variability in land-atmosphere interactions attributed to? |
|---|---|
| ridge | In multiple linear regression, attribution depends on the collinearity of covariates (e.g., circulation) with the second-order effect (land–atmosphere interaction), determining whether it is assigned to GMST or circulation changes. This partitioning may vary by region. If there is no strong collinearity between the prevailing atmospheric circulation at the daily time scale and land-atmosphere interactions, which typically change at longer time scales (Merrifield et al., 2017), we expect the effects due to land-atmosphere interactions to remain in the residuals. |
| analogues | It is presumed that circulation analogues occur over a range of land surface states. The circulation-induced component of temperature is defined as an average across this range, which leaves the influence of the land surface on the atmosphere predominantly in the residual thermodynamic component (Merrifield et al., 2017). The land surface can induce a temperature anomaly and associated circulation pattern (e.g., a thermal low), and the analogue method could interpret this situation as circulation-induced rather than thermodynamic. Nudging all vertical levels of the atmosphere suppresses influence from the land surface to the atmosphere, so land-atmosphere interactions are likely to remain in the thermodynamic component of the piControl-nudged runs used as a benchmark in this study (Merrifield et al., 2019). |
| DEA | Because the approach removes the total effect of GMST without conditioning on land–atmosphere interactions, it may also eliminate the mediating effect of GMST operating through this pathway (but this effect is likely small and confined to trends that are colinear with GMST), whereas the mediating effect of atmospheric circulation is expected to be retained, given that the linear model has sufficient expressive capacity to capture these complex relationships. |
| UNET | The SLP is used as a predictor of the circulation. However, this variable may contain surface imprints which might affect the "circulation-induced" component. Therefore, land-atmosphere interactions may be partly predicted by the UNET architecture used here. |
| nudged simulations | Nudging is expected to separate the land-atmosphere interactions into a thermodynamically-driven, and a circulation-driven component (i.e., atmospheric imprints of land-atmosphere interactions are expected to be captured through nudging). However, the circulation patterns that are extracted from the forced transient simulations, contain thermodynamic component due to land-atmosphere coupling at the planetary boundary layer. |

Regarding land-atmosphere interactions, we conclude that the effect on our estimates of circulation-induced trends varies between methods. This increases our confidence in the signals all methods agree on (e.g., circulation-induced warming over Europe). At the same time, there is no systematic (and consistent) difference in how statistical decomposition methods might be affected by land-atmosphere interactions in comparison to how land-atmosphere interactions might influence the nudged simulations. Therefore, the effect of land-atmosphere interactions cannot explain the systematic underestimation of the magnitude of circulation-induced trends in statistical decomposition methods (as compared to the nudged simulations).

Fourth, we present an evaluation of decomposition methods based on one set of nudged simulations from one Earth system model (CESM2). Despite the well-documented performance of CESM2, this is a flaw, as the strength of the links between atmospheric circulation patterns, GMST, and local temperatures might be misrepresented in the model. A follow-up study using multiple ESMs to create a benchmarking dataset would be crucial to constrain our estimates of circulation-induced temperature trends further. The use of such a multi-model ensemble would require adapting the different reconstruction methods slightly. In particular, the UNET would need to be pre-trained on a collection of multi-model data, rather than just CESM2. Preliminary tests conducted on Western Europe suggest that this does not degrade the quality of the reconstruction, especially when the fine-tuning step is applied to early ERA5 data.

Finally, in addition to combining multiple lines of evidence, our study emphasizes the importance of benchmarking efforts for statistical and machine learning approaches. Without evaluating the results against nudged circulation simulations, one would conclude that different decomposition methods project similar trend patterns, with some estimates exhibiting a stronger version of the trend pattern than others. Evaluating which magnitude of the trend pattern is the most likely/plausible is challenging from the statistical analysis alone. Concluding that all decomposition methods applied to observations might be underestimating the magnitude of the trend pattern would be impossible.

Overall, our study demonstrates that statistical methods can effectively identify and separate circulation-induced temperature trends from residual thermodynamic trends. However, their performance declines on climatic timescales compared to shorter timescales. This uncertainty should be carefully considered in future studies that use such estimates for attribution or to constrain projections.

## 5  Conclusions and Outlook

In summary, our analysis targeted two specific research objectives and revealed two distinct findings: First, we evaluated whether statistical-empirical methods can accurately estimate circulation-induced long-term trends in the NH mid-latitudes during boreal summer (and a residual dominated by thermodynamic trends) against a specifically designed climate model benchmark of nudged circulation simulations. Four different statistical methods were tested, and we demonstrated that each of these methods can generally identify the large-scale pattern of circulation variability and changes, even though they are typically trained and validated on short time scales (daily to seasonal). However, the methods showed differences in their ability to reproduce the spatial trend pattern from the nudged circulation benchmark. With three-quarters of correctly estimated signs of trends and coefficients of determination above 50%, the ridge regression and the UNET methods are performing

sufficiently well for the purpose. The UNET has the overall highest scores in most tested skill metrics. However, the UNET method tends to produce underdispersive results, that is, the magnitude of particularly strong circulation trends is often underestimated (irrespective of the sign). DEA and circulation analogues have similar skill in predicting the sign of circulation-induced trends. Still, due to the low coefficient of determination, we would refrain from interpreting the magnitude of regional trends estimated from these methods. Overall, identifying circulation-induced trends on climate time scales in the context of dynamical adjustment studies is possible. Still, it does imply larger uncertainties than for the application on shorter time scales, which needs to be considered in future applications of the techniques.

Our second objective was to identify circulation-induced boreal summer temperature trends across the northern mid-latitudes using four statistical methods and CESM2 simulations nudged to ERA5 circulation without anthropogenic forcing. Large-scale summer circulation trends and their temperature impacts have been widely debated (Teng et al., 2022; Chemke and Coumou, 2024; Rousi et al., 2022; Vautard et al., 2023). Analyzing ERA5 for 1979–2023, we find positive circulation contributions to summer warming over Europe, western North America, and Mongolia. In contrast, a wave-like pattern of circulation-induced cooling appears over Central Eurasia (west Siberia and Kazakhstan) and Central to Eastern North America.

Beyond strengthening confidence in circulation-induced temperature changes, our evaluation also highlights systematic limitations of statistical decomposition methods. Improving understanding of their performance will enhance our ability to attribute regional climate trends. Isolating individual components of historical change is likely to yield a stronger attribution signal, particularly for regional climate change. Focusing on dynamical and thermodynamic changes separately is advantageous, as there are significant differences in the uncertainties of forced changes in these components (Shepherd, 2014). While several attribution studies of circulation changes have been published (Coumou et al., 2015; Chemke and Coumou, 2024; Dong et al., 2022), uncertainties remain large, especially when it comes to attributing the downstream impacts of atmospheric circulation changes. Being able to more robustly decompose a trend into a circulation-induced and a thermodynamic component should also help attribute circulation-induced temperature trends more effectively.

Finally, separating dynamical and thermodynamic components offers a pathway to constrain near-term climate projections using observation-based constraints. The thermodynamic constraint should be straightforward to identify, as it is mainly forced. There are different possibilities to constrain based on the dynamical component. With the assumption that circulation-induced trends over the past decades were primarily due to internal climate variability, one would expect a reversal of the observed trend pattern over the coming decades. If circulation-induced temperature change is forced, both circulation-induced and thermodynamic trends would continue. Due to the considerable uncertainty in the forced circulation-induced changes, a storyline approach would be appropriate, explicitly treating different assumptions about forced atmospheric circulation changes and evaluating the potential outcomes of these scenarios (Shepherd, 2019; Liné et al., 2024).

. The code required to reproduce this study is available on https://github.com/peterpeterp/circ_contribution_to_JJA_trends.git (Pfleiderer, 2025).

## Appendix A: Nudged circulation plots

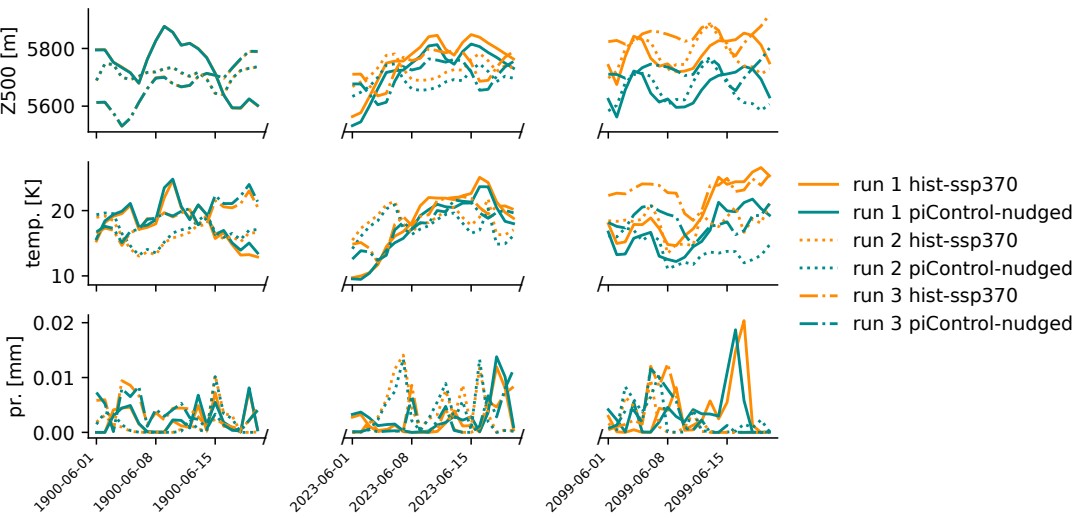

**Figure A1.** Differences in temperature, precipitation and geopotential height at 500hPa at one grid-cell (next to Leipzig). Orange lines represent hist+ssp370 simulations, green lines represent the corresponding piControl-nudged simulations.

## Appendix B: Significance of circulation induced trends

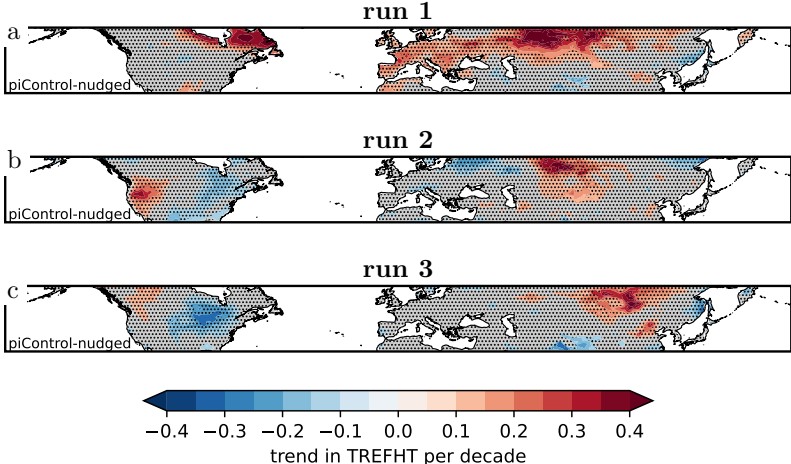

**Figure B1.** JJA trends in piControl-nudged simulations for the period 1979-2023. The Stippling indicates that we cannot reject the Null-hypothesis of no trend at a 95% level.

## 455 Appendix C: More evaluation plots

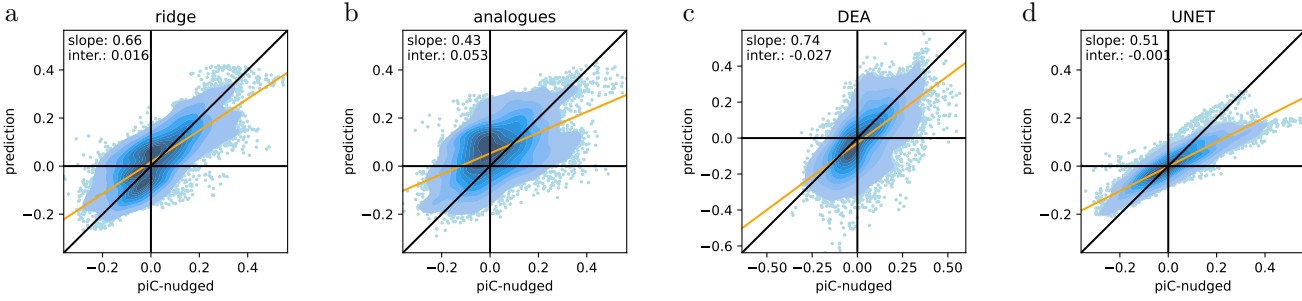

**Figure C1.** Predicted trends against trends from piControl-nudged for the period 1979-2023 over land grid-cells between 30N and 60N from all runs (1,2,3) for the ridge regression (a), DEA (b), analogues (c), and UNET (d). The bulk of the data is represented by a Gaussian kernel density estimate (shadings) while extreme trends are shown as scatter points.

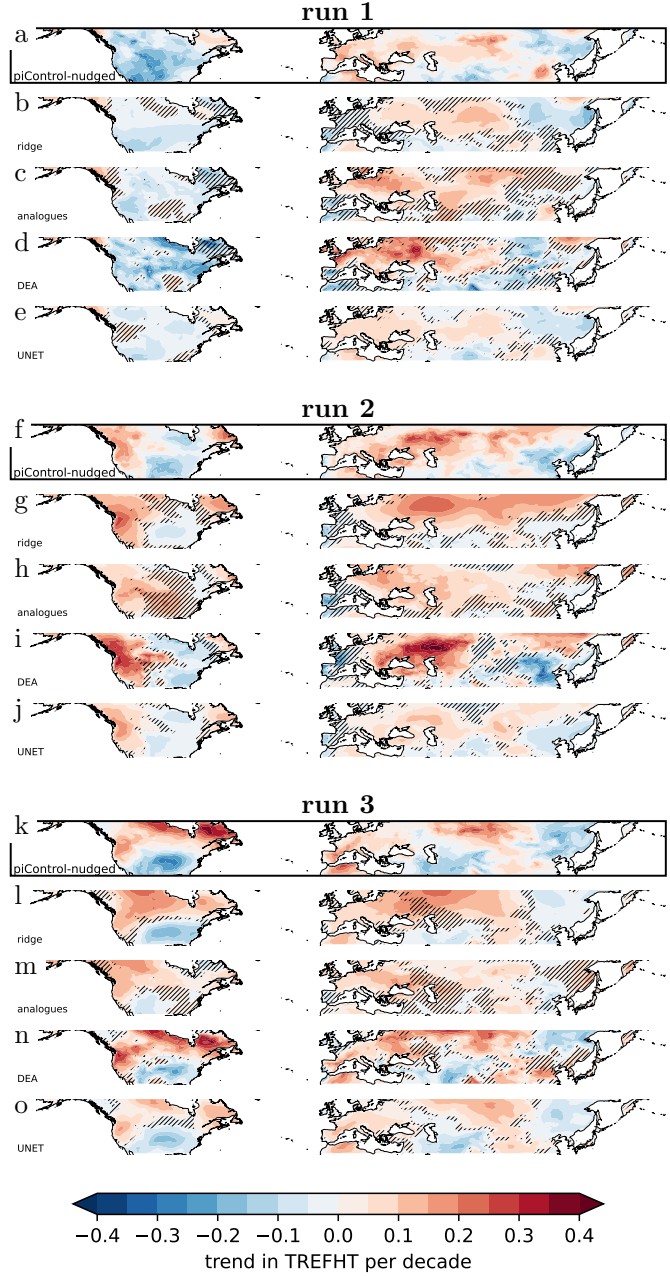

**Figure D1.** Same as Figure 3 but for the period 2025-2075.

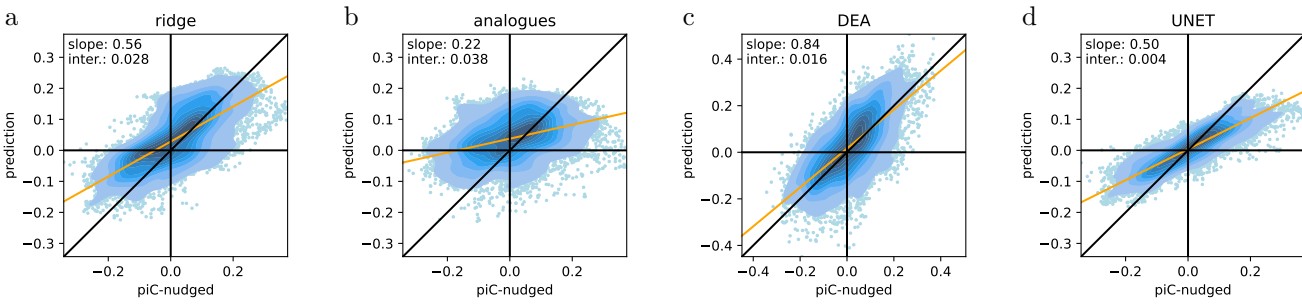

**Figure D2.** Same as Figure C1 but for the period 2025-2075.

**Table D1.** Same as table 1 but for the period 2025-2075.

| | ridge | analogues | DEA | UNET |
|---|---|---|---|---|
| **run** | | | | |
| *correct sign* | | | | |
| **all runs** | **77%** | **65%** | **78%** | **87%** |
| run 1 | 74% | 63% | 75% | 87% |
| 1400 | 77% | 68% | 78% | 85% |
| run 3 | 81% | 64% | 81% | 89% |
| *Pearson correlation (r)* | | | | |
| **all runs** | **0.71** | **0.37** | **0.70** | **0.86** |
| run 1 | 0.64 | 0.44 | 0.56 | 0.82 |
| run 2 | 0.72 | 0.31 | 0.73 | 0.83 |
| run 3 | 0.73 | 0.29 | 0.79 | 0.91 |
| *coefficient of determination (R2)* | | | | |
| **all runs** | **0.44** | **-0.00** | **0.22** | **0.66** |
| run 1 | 0.40 | -0.09 | -0.20 | 0.58 |
| run 2 | 0.35 | -0.07 | 0.10 | 0.60 |
| run 3 | 0.47 | 0.00 | 0.52 | 0.73 |
| *regression slope* | | | | |
| **all runs** | **0.56** | **0.22** | **0.84** | **0.50** |
| run 1 | 0.43 | 0.33 | 0.71 | 0.47 |
| run 2 | 0.60 | 0.20 | 0.99 | 0.48 |
| run 3 | 0.53 | 0.13 | 0.76 | 0.54 |

## Appendix E: UNET architecture

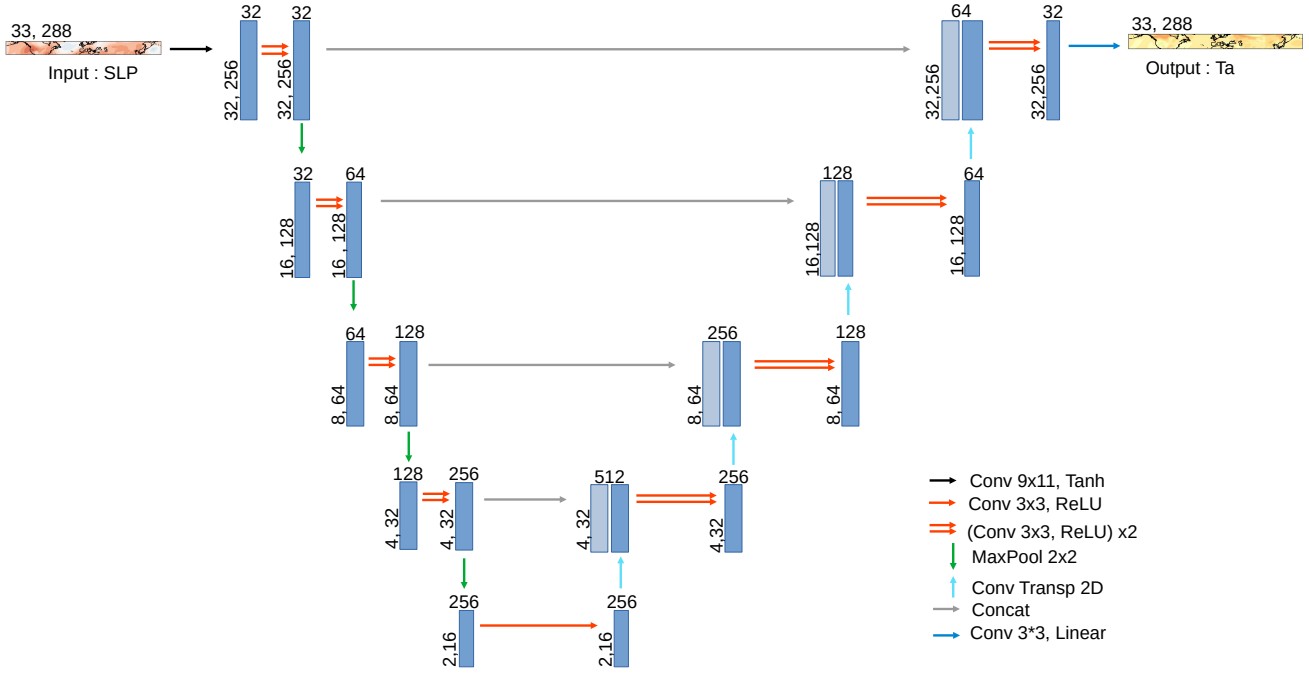

**Figure E1.** UNET architecture.

## Appendix F: Analogues stochastic

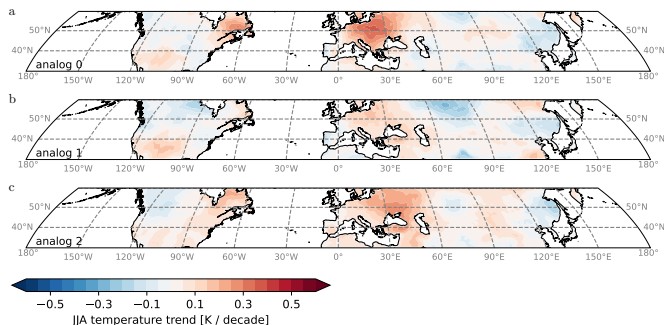

**Figure F1.** Circulation-induced contribution to JJA mean temperature trends in ERA5 over the period 1979-2023 estimated by the analogue method. ERA5 is dynamically adjusted using analogues selected from (a) run 1, (b) run 2, and (c) run 3.

## Appendix G: Ridge regression with geopotential heigt at 500hPa

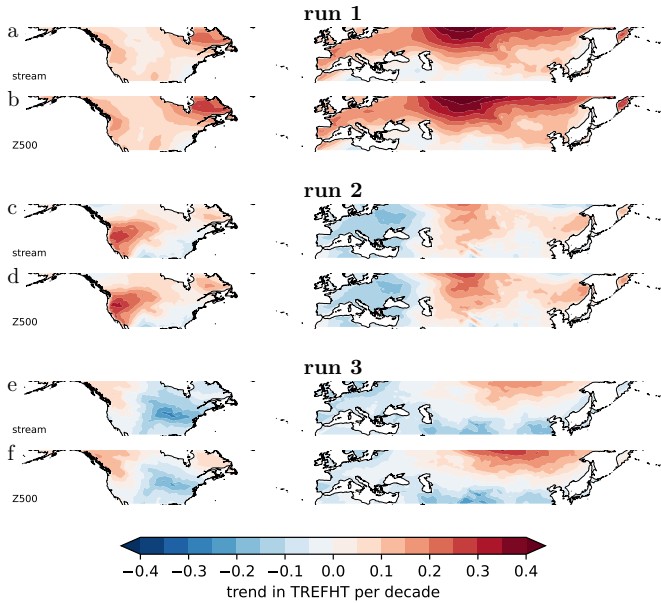

**Figure G1.** Estimates of the circulation induced trends from the ridge regression over the period 1979-2023 in the freely running forced CESM2 simulations. Estimates from the ridge regression using streamfunction at 500 hPa as a covariate for circulation (a,c,e). The same, but with geopotential height at 500 hPa as anomalies to the global mean geopotential height at 500 hPa (b, d, f).

. PP and SS conceived the study. PP wrote the manuscript with contributions from SS and all other authors. PP created all figures. PP contributed the results of the ridge regression. AM contributed the results for the constructed analogues. HD contributed the results for the DEA. EC and JC contributed the results for UNET. ID contributed the CESM2 simulations nudged to ERA5 winds.

. No competing interests are present.

. We thank Urs Beyerle for producing the nudged CESM2 simulations that were used to evaluate the statistical decomposition methods.
We thank all the scientists, software engineers, and administrators who contributed to the development of CESM. We acknowledge the use of ERA5 reanalysis data provided by the Copernicus Climate Change Service (C3S). We acknowledge the CESM2 Large Ensemble Community Project for providing the model output used in this study. S.S. and P.P. acknowledge the project 'Artificial Intelligence for Enhanced Representation of Processes and Extremes in Earth System Models' (AI4PEX; grant agreement 101137682), funded by the EU's Horizon Europe program; and the climXtreme project funded by the German Federal Ministry of Education and Research (Phase 2, project
PATTETA, grant number 01LP2323C). S.S. and I.D. acknowledge funding from the German Research Foundation's Heinz Maier-Leibnitz Prize 2024 for early-career researchers.

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
