# Peer review of "Considerable yet contrasting regional imprint of circulation change on summer temperature trends across the Northern hemisphere mid-latitudes"

_EGUsphere, 2025_

## Editor Comment (EC1)

**EGU peer-review training 2025**

Review of "The contribution of circulation changes to summer temperature trends in the northern hemisphere mid-latitudes: A multi-method quantification" by Pfleiderer et al. (egusphere-2025-2397)

**Review by Arundhati Kalyan, Anjali Thomas, and Jan Zibell\**

\*J.Z. declares being employed at the same institute as one of the co-authors of the study.

In the above study, the authors assess long-term (multidecadal) circulation-induced changes in summer temperature trends in the northern hemisphere midlatitudes (30–60°N). They employ four different statistical/machine learning-based methods – ridge regression, atmospheric circulation analogues, direct effect analysis, and a convolutional neural network – to isolate temperature trends driven by circulation changes over historical and future time periods in freerunning climate model simulations (using CESM2) and ERA5 reanalysis. The relative performance of these methods is evaluated against a benchmark comprising nudged climate model experiments (also using CESM2) which include forced and internal components of circulation variability, but no forced thermodynamic component. The four methods are found to be effective on climate timescales, albeit with biases. The paper also highlights regional differences in dynamically forced trends, revealing alternating wavelike patterns of warming and cooling throughout North America and Eurasia. Finally, the authors discuss in depth the challenges and limitations in using statistical methods to decompose circulation vs. thermodynamically driven trend signals.

The study makes robust choices relating to data and methodology, in that four different decomposition methods are used and multiple statistical evaluation metrics are analysed for each. The authors transparently present a summary of the climate of the wind-nudged simulations which form the reference for their assessment of the decomposition methods. The presentation of challenges (multidecadal timescales, differing climate components included in the circulationrelated decomposition, creation of a suitable benchmark, need for multiple models) in the discussion section is strong and may serve as useful reference for future studies. The study completes the stated target of quantifying circulation-driven trends across the NH midlatitudes and validating the different methods against a suitable climate model benchmark. In addition to highlighting features with low and high skill for each method, the study adds some degree of confidence to the findings of earlier studies that examined regional patterns of circulation-driven temperature trends. The article is concise and generally well-written. The title is informative and appropriate to the study, although the abstract could be improved as suggested below. The relevance of the study, objectives, and scientific challenges are introduced well. Even so, there are portions of the manuscript that are hard to follow and hinder conceptual understanding and interpretation of the results. In particular, the explanations of the statistical methods, the illustration of statistical significance/uncertainties associated with the results, and the dynamical interpretation of the study could be improved. We recommend publication of this manuscript in Weather and Climate Dynamics after minor but necessary revisions as outlined in the comments below.

**Main comments:**

- Statistical methods: The description of the four methods used is not straightforward and quite hard to follow for a non-expert reader. We are not experts in the statistical/ML methods used in this study and hence, cannot comment on the strengths and weaknesses of the design and implementation of these methods. However, here are a few suggestions to make the methods understandable to a broader dynamics audience, such that we can better appreciate the significance of the study's findings:
  - Add an introductory sentence or two in plain language in all of the sections 2.3.1–
     2.3.4 to explain what the method and/or equation aims to achieve
  - o Clearly define all variables and constants introduced in each equation
  - Consider also whether certain details in the method descriptions can be moved to supplementary text
  - Here a few line-specific comments:
    - Line 134: Please explain why a 40°×40° rectangle around each grid cell was chosen. Could other sizes change the results?
    - Line 156: Please discuss briefly how you choose the number of analogues (Ns = 50) and repetitions (Nr = 100). Does changing these choices affect the results?
    - Line 200: It might help to briefly explain the physical meaning of the Yperp component in plain language in this context.
- It would be good to see more rigorous discussion of the statistical significance and uncertainties associated with the results presented. The need for this arises due to the following reasons:
  - o "... observed trends are falling out of the range of model-simulated expected trends" (Line 35)
  - There is "ambiguity" on the magnitude of the historical circulation-induced trends, and all methods likely underestimate these magnitudes (Sec. 3.2)
  - o In light of the above, it would be helpful to better understand the likely range/distribution of trends for each method and how they differ. Are some methods more likely to include the observed/nudged values than others?
  - The use of a very small sample size (only three ensemble members) for the freerunning and nudged simulations limits the ability to show robust uncertainties. Could you provide more information on whether the initial conditions for the three nudged ensemble members were chosen at random or based on certain criteria? Could you comment on how large of an added value more members would be and why you decided on three only?
  - Table 1 would also benefit from the inclusion of confidence intervals and/or p-values to reject the null hypothesis that the correct sign or correlation of temperature trends was obtained by random chance.
- There are instances where we think that the discussions in this study may benefit from a more dynamical perspective:
  - o It is clear from the introduction and methods that the wind-nudged simulations are viewed as a ground-truth benchmark. The limitations of this approach are

- discussed as, for instance, in line 335: "However, there may be factors of residual climate variability (such as ocean variability) or feedbacks between circulation and other factors such as land-atmosphere coupling that could still affect thermodynamical processes on climate over land." Aren't there even more concrete examples for a physical relationship that is not a priori captured, such as the time mean thermal wind balance?
- o Line 36–37: Please consider to slightly reframe "... may indicate ... that a forced change in circulation is missing in the models". This framing could make one think of unresolved/parameterized processes in climate models, such as latent heating in deep convection in the midlatitudes. For those it is the consensus that a forced circulation change *is* missing in the models. Should you actually refer to this, the sentence could be reframed as "that circulation changes due to unresolved processes, which are not captured by the models, turn out to be meaningful / non-negligible".
- O It is certainly not the purpose of this study to discuss all limitations of windnudging in detail and of course every alternative also has its limitations, but we
  suggest that the authors at least address the point that in the real atmosphere, the
  development of a heatwave is not the linear sum of a thermodynamic component
  plus a dynamic component. There are many non-linear dynamical feedbacks from
  thermodynamical processes, e.g. from coupling with radiation via clouds, surface
  fluxes depending on soil moisture, or latent heat release (which, if speaking of
  heat waves, within a warm conveyor belt may increase blocking intensity (Pfahl
  et al. 2015)). Possibly, addressing this is related to emphasizing more prominently
  that the authors regard GMST as the representative variable of thermodynamic
  changes.
- Overall, we suggest that 1) the assumption that a trend can be decomposed into thermodynamical and dynamical components and 2) the use of wind-nudged simulations (as introduced in Sect. 2.1) to achieve this are discussed a bit more critically. This could be done very briefly when introducing the wind-nudging experiments in the Introduction and then in a more elaborate way for instance as a sixth discussion point in Sect. 3.3.
- The main objective of the study is to investigate long-term temperature trends. Therefore, the prominent motivation based on heatwaves and extremes seems somewhat out of place. Motivating this research with individual events is fine per se, but this study does not investigate trends in heatwaves. It would be helpful if the authors could discuss a more concrete example of how their estimated trends allow a better understanding of extreme events (as indicated in lines 31-34)?
- The use of the dynamical vs. thermodynamical separation of trends could be even further strengthened by a discussion of the geographical variability of the thermodynamically induced trends. In Figure 4, over Eurasia your methods disagree whether the thermodynamical change is rather uniform or dependent on longitude or latitude. Are there any physical arguments in the literature for what the thermodynamically-induced pattern of warming should look like? For instance, it is observed that the Mediterranean

region is warming faster (Brogli et al., 2019). Alternatively, it seems also fine to note that this is left for future study or refer to discussions in other studies.

**Minor comments:**

- Line 3–4: "Over the northern hemispheric mid-latitudes, considerable regional differences in summer temperatures have been observed." → Presumably, you mean differences in summer temperature *changes*.
- Line 5–6: We think the general readability of the abstract would benefit from a brief description of 'decomposition method', i.e., what you decompose the trends into. If one is not very familiar with the topic or similar literature, this is not obvious but only (and well) presented in the introduction.
- Line 10–11: "Most decomposition methods show skill in estimating the sign of circulation-induced trends but all methods underestimate the magnitude of these trends." This statement contains the fact that you use the wind-nudged simulations as your benchmark and that you assume that the nudged simulations contain 100% of the dynamical component of the trend. This should be presented more clearly as was done in the Introduction, for instance, around line 60.
- Line 16: Consider changing: The intensity of heatwaves "increases globally" to "has been increasing globally".
- Line 18–19: Consider adding a reference/s here to show that intensification of heat waves occurs in a warmer climate due to thermodynamic factors (perhaps an attribution study?).
- Line 19: "However, heat waves are not only..." is a long sentence and could benefit from restructuring.
- Line 22: It might be worth mentioning here whether land-atmosphere interactions are more or less important than circulation changes as a factor in driving summer temperature trends.
- Line 23: It might be worth commenting on whether regional trends are more pronounced in the NH midlatitudes than elsewhere. May also be good to include a sentence citing studies that examined trends in the tropics or Southern Hemisphere.
- Line 24–27: This is a long sentence and could be split into two separate sentences.
- Line 31: Consider including more specific references that show why forced changes in circulation are small compared to internal variability.
- Line 48: Consider whether there are any other limitations of the nudged circulation experiments and include that here.
- Line 49: Add "e.g. the circulation not being in thermal wind balance" to clarify that you don't mean unresolved processes, which are also mechanisms not represented in the models used.
- Line 49–51: On the other hand, most of statistical decomposition methods ..." can be rewritten/shortened or split into two sentences to enhance readability.
- Line 54–56: "Moreover, benchmarks for circulation-induced long-term trends have not been available so far, and to our knowledge no systematic comparison of dynamical adjustment methods has been performed." It would be good to clarify if this applies globally or just to mid-latitudes and to briefly mention if any emerging efforts exist. This would make it clearer why the study is filling an important gap.

- Line 57: Add the specific NH latitude range being examined here.
- Section 2: Adding a sentence or two linking each data/methods subsection to the main aim (separating thermodynamic vs circulation-driven temperature trends) would help the reader understand why each method or simulation is being used.
- Line 69: You could specify the ERA5 years here as well.
- Line 72: "First, three standard historical and future forcing experiments ....". At first reading, this sounds like three different forcing scenarios or the like. Please specify that you mean three ensemble members.
- Line 80: How about specifying some of the main features of the nudging, e.g., whether your nudging is done at the model grid or involves some spectral transformations, and to which vertical level it is done? This way the reader gets a good first impression without having to refer to Topal and Ding (2023) to find out what is "similar" to their approach and what is different.
- Line 80: Is CAM6 an abbreviation of something? If yes, please mention.
- Line 81: "These simulations will be henceforth referred...". Simplify this sentence for readability.
- Line 87: The phrasing of thermodynamic forcing being represented by "surface temperature" somewhat suggests that using this metric of forcing is not a choice. In reality, temperature change is non-uniform throughout the atmosphere with implications for the midlatitude circulation. We suggest that the authors instead say 'commonly approximated by GMST' or similar.
- Line 89: Good point, but it would help to add a short explanation of why it is hard to evaluate these methods in a coupled system.
- Line 95: Will the residual internal variability (e.g. from the ocean) influence the evaluation of the decomposition methods, and how do you account for that?
- Line 111: "However, we assume that the effect...": This assumption is reasonable, but you might want to add a reference or a short justification for this assumption.
- Figure 2: It is not clear what is meant with the cooler versus warmer histograms in panels b), c). Please clarify.
- Line 115: Consider splitting the paragraph into two shorter ones: one describing the experimental setup and another explaining its implications and limitations. This would improve readability.
- Line 120: Briefly define AMIP in the text for clarity.
- Line 132: Could split into two shorter sentences for clarity.
- Line 216: Consider providing more details of the transient CESM2 simulations used to train the UNET.
- Lines 219–220: Clarify why the training is done on CESM2 first and then fine-tuning on ERA5—why does this improve performance or robustness?
- Line 220–222: Rephrase to sound more concise and formal.
- Line 227–237: Consider presenting these two sets of bullet points together instead of separately to make the text more concise and easier for the reader to associate each skill metric with what it represents.

- Section 3.1: The discussion mentions how the methods differ (DEA captures magnitude, UNET conservative), but the rationale behind these differences could be explained more clearly. For instance, why does UNET underestimate magnitudes?
- Line 276: This is a long sentence. Consider splitting it into 2–3 smaller sentences to enhance readability.
- Line 289–291: In "The ridge regression ... up to 0.6 K/dec", change "where" to "were", and add "*suggest* stronger circulation induced trends ..."
- Fig. 4: Which wavelength could approximate the wave-pattern change that you find? Can you relate this to other studies?
- Line 305: This paragraph sounds like a re-introduction of dynamical adjustment from zero. A bit of repetition is appreciated for the flow, but at the current stage this introduction of dynamical adjustment is even clearer than in the introduction (using even more references). Please consider streamlining this or, otherwise, stating more clearly if you mean something different than in the introduction or possibly moving some of this material into the introduction.
- Section 3.3 is purely a discussion. Why not make it a new section called Discussion? There is no new result in this section.

**Technical comments:**

- Line 35–36 and onward: Check for the use of citet vs. citep and citep[][]{} throughout the paper.
- In Figure 3, the kernel density maps could be enlarged with axes labels shown.
- Figure A2 is not explained or referenced in the text. Change.
- Figure B2 is not explained or referenced in the text. Change.

**References:**

Brogli, R., N. Kröner, S. L. Sørland, D. Lüthi, and C. Schär, 2019: The Role of Hadley Circulation and Lapse-Rate Changes for the Future European Summer Climate. J. Climate, 32, 385–404, https://doi.org/10.1175/JCLI-D-18-0431.1.

Pfahl, S., Schwierz, C., Croci-Maspoli, M. et al. Importance of latent heat release in ascending air streams for atmospheric blocking. Nature Geosci 8, 610–614 (2015). https://doi.org/10.1038/ngeo2487

---

## Author Comment (AC1)

**Overall review**

This paper by Pfleiderer et al. aims to improve our ability to decompose climate trends into thermodynamic and dynamical components, with a focus on surface temperature trends in the Northern Hemisphere. The first one is to determine whether statistical methods are able to quantify dynamically induced trends in climate model data by comparing their outcomes to a set of nudged climate models experiment, considered to be the ground truth. Once this is validated, the statistical methods and another set of nudged experiments are applied to ERA5 data to actually determine the contribution of dynamical changes to the surface temperature trends in the northern mid-latitudes.

The paper is highly relevant and timely, and it provides an important assessment of dynamical adjustment techniques. Beyond its specific results, the framework developed could be applied to a wider range of climate variables, such as precipitation or extreme events.

The use of nudged simulations with no external forcing is a particularly smart approach to isolate the dynamical influence on surface temperature trends. Since such experiments are difficult to construct for observational datasets (though the AMIP + nudging above 700 hPa approach seems promising), validating statistical methods is crucial, and this paper does so effectively.

The manuscript is generally well written, although it can be hard to follow at times. Some sections, particularly on the analogues method, would benefit from clearer explanations. Also, the two main objectives, though related, are presented somewhat independently and could be more tightly connected in the structure of the paper. For example the authors could emphasize that the first objective is used to strengthen our confidence in the second objective.

Despite some concerns I have about the paper (detailes below), I think this paper is almost suitable for publication in WCD, but requires some work, notably to improve clarity. For these reasons I suggest to **accept this paper with minor revisions**.

*Thanks for the positive feedback and for pointing out parts of the manuscript that can be improved.*

**Main comments**

**1. Comparability between methods**

One of my main concerns is the comparibility between the different statistical methods. Indeed, each method uses a different set of predictor variables:

- Ridge regression uses the streamfunction
- Circulation analogues use sea-level pressure
- Direct effect analysis and U-Net use z500

This makes it difficult to assess whether differences in performance are due to the method itself or the choice of input variables. It would be helpful for the authors to comment on this explicitly. If the best predictor was chosen for each method, this should be clarified.

*Our aim with this article is to evaluate how reliable statistical and machine learning methods for trend decomposition are. We did not develop these methods for the task of disentangling circulation induced changes from thermodynamic changes but rather used already existing methods that are likely to be used for the task. Therefore, we prefer applying the methods as they were used beforehand in the published scientific literature, which entails the usage of different proxies for atmospheric circulation.*

*For most of the methods, sensitivity tests with other input variables representing atmospheric circulation showed that the choice of the input variable does not impact the result considerably.*

*In the revised manuscript we compare results with the ridge regression with streamfunction at 500 hPa as input variable to results from the same ridge regression but with geopotential height at 500 hPa (corrected by subtracting the global mean of geopotential height) as input (see figure G1). The results are very similar:*

[Figure]

*Figure G1. Estimates of the circulation induced trends from the ridge regression over the period 1979-2023 in the freely running forced CESM2 simulations. Estimates from the ridge regression using streamfunction at 500 hPa as a covariate for circulation (a,c,e). The same, but with geopotential height at 500 hPa as anomalies to the global mean geopotential height at 500 hPa (b, d, f).*

*In the revised manuscript, we inform the reader about this aspect in line 139-142:*

*"Note that we apply the methods exactly as they were designed and used in other publications and therefore different proxies for atmospheric circulation are used by different methods. We do not expect that the choice of the variable to represent atmospheric circulation affects the results considerably. In figure G1 we show a sensitivity analysis for the ridge regression. In section 3.3 we discuss differences between the methods and how they might affect the decomposition in more detail."*

**2. Lack of information on trend estimation, significance and uncertainty**

Maybe I have missed it but I couldn't find a mention on how the trends were computed. In addition, such a study would benefit from statistical tests on trend significance and uncertainty, especially for the second objective which aims to provide robust estimates. As all methods provide an estimate of surface temperature directly, trends statistics could be computed for all cases. Moreover, it might make sense to evaluate skill metrics only for statistically significant trends.

*We agree that a discussion on the significance of analyzed trends is lacking. Circulation induced trends are weak compared to thermodynamic trends. To which extent anthropogenically forced changes in atmospheric circulation patterns is subject of debate. It is however clear that a large part of circulation induced trends over a time period of 45 years is a result of internal climate variability. The differences between the nudged piControl simulations (figure 3 in the original manuscript) suggest that in CESM2 most of the circulation induced trends at a local scale mostly reflect internal climate variability. Whether this would be the same in other climate models or in observations is a question we do not address here. Either way, we assume that a large part of the circulation induced trends is driven by internal climate variability and therefore we expect that most individual circulation induced trends are not statistically significant (yet for the thermodynamical trends we expect much less variability and therefore a high proportion of significant trends).*

*The consistent regional patterns we find in the circulation induced trend maps show that the trends are the result of processes in the climate system and we want to quantify these contributions even though from a statistical point of view some individual trends are not statistically significant.*

*In the revised manuscript, we show maps with significance stippling in the appendix (figure B1):*

[Figure]

*Figure B1. JJA trends in piControl-nudged simulations for the period 1979-2023. The Stippling indicates that we cannot reject the Null-hypothesis of no trend at a 95% level.*

*We also add a paragraph on the significance of the trends in the results section (line 267-270):*

*"Note that most of these trends in the atmospheric circulation-induced component are not statistically significant (see figure B1). Since these trends mostly reflect internal climate variability, it is expected that from a statistical point of view, the circulation-induced temperature changes at one location are not differentiable from noise. The spatially consistent trend patterns show that, despite lacking statistical significance, these trends contain helpful information and are worth evaluating."*

**3. Section 2.3.2 (circulation analogues) lacks clarity**

The description of the analogue method is quite confusing. As someone who is not familiar with circulation analogues, I cannot say I have understood what it is from that section. Please revise it to make it clearer. Here are the points that made it unclear to me:

- "Analogues" are not clearly defined when first mentioned (line 155).

  *We've revised to define analogues at first mention. Thank you for the clarification!*

- It is unclear what the 80 possible choices for analogues refer to - days, years, months?

We've moved towards an example to emphasize the application of the method on monthly mean fields.

- What are these 50 out of 80 choices? I did not understand this paragraph

This detail is to orient readers familiar with applications of the method in previous papers. The step refers to the strategy of going from the whole record as possible analogues to a subset of the record as possible analogues. We have hopefully clarified that by re-ordering the paragraph and adding detail.

- Line 128: "Once the Euclidian distances are determined" at that point there is no indication that a Euclidian distance is computed, or why and on what it is computed.

Thank you for this point, we had gotten ahead of ourselves a little. We have moved the mention of Euclidean distances from the later paragraphs into the first paragraph so (hopefully) it is now clear what is being done.

- Line 169: analogues are now defined but this should be done earlier

  Done, thank you!

Maybe this is also the case for the UNET paragraph, but as I am more familiar with UNETs it was easier to follow.

**4. Are the UNET Predictions Truly Circulation-Induced Temperature Changes?**

- Line 213 describes the UNET model as predicting a temperature field with an estimate of the daily non-stationary normal removed. However, this doesn't necessarily isolate the circulation-induced component. The paper assumes that the resulting anomaly is circulation-induced, but this should be justified more clearly.
- If the previous point is justified (which I am sure it is) why not use the method from Rigal et al. (2019) directly to estimate circulation-induced temperature changes?
- Is the UNET performing well to reproduce this anomaly field?
- Why not use the UNET to predict the nudged experiment directly, which serves as the ground truth for the comparison later

Also, it is unclear what "CESM2 transient simulations" refers to. Do these include historical + SSP runs?

The aim of the UNET approach is precisely to estimate the part of daily temperature variations which can be explained by the large-scale circulation (here assessed from daily SLP fields). The mean seasonal cycle of the temperatures is not circulation-induced, so it is relevant to remove it and focus on temperature anomalies (T'): this is why we write the UNET model as T' = f(SLP).

The UNET is then trained to learn the link between SLP and T'. As we train the UNET on historical + SSP runs (which we call 'transient', i.e. non-stationary), we need to account for climate change in the T' = f(SLP) relationship. Here we detrend the temperatures but not the SLP, assuming that, in this model (CESM), the forced response in the SLP is small compared to the daily variability --- which seems to be a reasonable assumption as the 3 piControl-nudged experiments do not exhibit significant common trends (see Fig 2 and 3). The detrending is made following the method described by Rigal et al. --- estimation of daily non-stationary normals --- which is convenient as it allows us to remove both the mean seasonal cycle (first point) and the climate change signal at the same time.

**Minor comments**

- Table 1: I fail to understand how R2 values can be positive. Could you please explain?

  *We use the coefficient of determination "R2" for the evaluation of our results and in the revised method we clearly define it. It informs about how much of the variability in our benchmark for circulation induced trends is explained by the estimates from tested methods. It usually ranges from 0 to 1. Cases where it is negative indicate that just taking the mean of the data would perform better than using the tested model. This is the case in figure 3h for example. We discuss the interpretation of R2 in the paper (line 256-258):*

  *"(iii) The coefficient of determination (R2 ) is a widely used metric for spatial comparisons, as it accounts for the variance at each location and indicates how much of the observed variability is explained by the prediction. Yet, it is -in contrast to Pearson correlation- sensitive to any bias in the estimated average (Kvålseth, 1985); and hence it is possible that a statistical method shows a good spatial Pearson correlation in its estimates but a poor R2 score."*

- Figure 4: How are thermodynamic trends obtained? Are they estimated directly (e.g., from the ridge regression method) or as a residual (total trend minus dynamical trend)?

  *It depends on the method: In the ridge regression and DEA it can be directly estimated from the model. With the analogues and UNET it is the residual.*

- Line 185: To be consistent with the text, maybe consider using Y_orth instead of Y_perp

  *Done*

- Line 270: "to weak trends" should be corrected to "too weak trends".

  *Done*

---

## Author Comment (AC2)

Review of WCD paper "The contribution of circulation changes to summer temperature…" by. Pfleiderer et al.

Overall, the paper is well structured and well documents a thorough study on different methods of estimating the role of atmospheric circulation changes to trends in the northern midlatitudes.

My overall recommendation would be publication after some revisions, which are generally minor.

*Thanks for the positive feedback and for pointing out parts of the manuscript that can be improved.*

General remarks:

My main query with this paper is the interpretation of the nudged simulations as a "benchmark". This is an excellent approach to include but I'm not wholly convinced that this is necessarily a gold standard in term of attributing the changes due to circulation.

*We agree that the nudged simulations have their limitations and should not be seen as the "gold standard". In the revised manuscript we discuss these limitations in more detail and alert the reader about these limitations earlier in the manuscript by referring to the section on limitations.*

The nudging approach is elegant, and the demonstration in Figure 2 clearly shows that impact of the thermodynamic forcing on the global scale. However, on smaller scales I am less sure that the nudging strictly represents the contemporaneous circulation driven anomalies. A couple of conceptual examples of the potential issues are as follows:

1. In Figure 3 the nudged anomalies are consistently higher than those predicted by the individual methods over the Eurasian continent in the summer. The nudging constrains all seasons (not just summer) so there are likely to be other factors that are modified by the nudging that contribute to these – particularly soil moisture but also other factors such as vegetation, snow melt etc.. These depend on seasons preceding the summer in question. In general these are small but there is the potential for these to have a local influence over time that systematically enhances the temperature response. I suppose these can be summarised as being model "feedbacks" (from other seasons and any associated integrated response) that are explicitly not captured by any of the statistical estimates but are implicitly included in the nudged "benchmark".

*We thank the reviewer for this interesting thought. We did not discuss this aspect so far and are happily adding it to the revised manuscript (line 364-368).*

*"Additionally, summer temperatures in the nudged circulation simulations might be affected by nudging in other seasons. For example, circulation changes can influence soil moisture in late spring which would then have an impact on summer temperatures. This information is not used*

*by statistical decomposition methods. Consequently, we have to admit that the nudged simulations are not a perfect benchmark. Further analysis is required to understand how these limitations affect our estimates of circulation induced trends and whether a better suited benchmark test could be designed."*

*We think that this influence of nudging in the other seasons on our benchmarking test is limited. By nudging the circulation over a longer time we assure that the conditions at the beginning of summer are very similar in terms of SST patterns, soil-moisture and other important pre-conditioning drivers for summer heat. The differences in the starting conditions in early summer between the freely running forced simulation and the nudged piControl simulation are mostly of thermodynamic nature. Concerning soil-moisture for example, with the nudging we assure that the amount of rain bringing storms that pass over a region of interest in spring is the same in both simulations. However, the amount of precipitation from these storms and the amount of evaporation is different in the piControl simulation. These differences in soil-moisture are relatively small. Nevertheless, we agree that the information about these differences are not accessible to our statistical models and could lead to a systematic mismatch between our estimates for circulation induced trends and piControl nudged simulations.*

2. The second point is regarding the nudging to winds in the lower troposphere – here the nudging is performed on short timescales and, despite only forcing winds, the dominance of thermal wind balance on synoptic scales means that the nudging will have an effective local temperature forcing. This adjustment will be fast but, for example, any thermodynamic feedbacks that occur between say the land and the atmosphere (for example the strengthening of an anticyclonic high over continents during summer) will show up as being due to the "circulation" when in reality there is a non-negligible impact from thermodynamics. These should not be as well captured by the statistical methods as the information and feedbacks are not directly included but must be elucidated from the data output. Of course, on global scales (i.e. in terms of GMST) this will have no impact but in terms of estimating the "circulation" contribution to local temperature changes, the thermodynamic contribution to thermal wind balance adjustment will be attributed to "circulation".

*We agree that nudging the winds in the lower troposphere interferes with small-scale thermodynamic feedbacks and that this has to be considered when interpreting the nudged piControl simulations. Experiments where the wind fields are only nudged from 700 hPa upwards are very similar to the simulations where the whole troposphere is nudged (as used here). We therefore think that these nudging effects do not considerably affect our analysis. In the revised manuscript we briefly discuss the issue.*

*We would also argue that since the piControl simulations are nudged over long times, this effect is limited. In our simulations, at a given location and time, land-atmosphere interactions should be similar in the freely running forced simulation and it's nudged piControl counterpart, the only difference being that due to thermodynamic changes the interaction might be slightly amplified or dampened. For the intensification of an anticyclonic high due to land-atmosphere feedbacks, only the (potential) intensification of this feedback due to thermodynamic effects would be missing in the statistical estimates.*

*The raised concern points to an issue that we already discuss in the manuscript which is that the decomposition into "circulation-induced" and "thermodynamic" as we use it here is not very clean and different tested statistical methods treat this decomposition slightly differently. For instance, whether changes in land-atmosphere interaction are part of the "circulation-induced" or the "thermodynamic" contribution differs between the methods. This was a subject of discussion within the author team. Finally, we agreed that despite these differences in the methods, their results are commonly interpreted in similar ways and therefore it makes sense to allow for this inconsistency in the scope of the methods for our comparison.*

*We extended this part of the discussion adding a schematic figure (figure 5) and a table (table 2) where we summarize the expected implicit treatment of land-atmosphere interactions in the different methods.*

[Figure]

**Figure 5.** Conceptual illustration of causal relationships influencing local temperatures in a forced climate as in figure 1 but with an additional driver of local temperature. In this study we aim at decomposing the influence on local temperature into thermodynamic (GMST) and circulation induced contributions (in blue). Land-atmosphere interactions is a driver we do not explicitly model (in orange).

**Table 2.** Expectations on how land-atmosphere interactions might influence the decomposition into "circulation-induced" and "thermodynamic" contributions.

| method | what is variability in land-atmosphere interactions attributed to? |
|---|---|
| ridge | In multiple linear regression, attribution depends on the collinearity of covariates (e.g., circulation) with the second-order effect (land–atmosphere interaction), determining whether it is assigned to GMST or circulation changes. This partitioning may vary by region. If there is no strong collinearity between the prevailing atmospheric circulation at the daily time scale and land-atmosphere interactions, which are typically changing at longer time scales (Merrifield et al., 2017), we expect that effects due to land-atmosphere interactions remain in the residuals. |
| analogues | It is presumed that circulation analogues occur over a range of land surface states. The circulation-induced component of temperature is defined as an average across this range, which leaves the influence of the land surface on the atmosphere predominantly in the residual thermodynamic component (Merrifield et al., 2017). It is possible for the land surface to induce a temperature anomaly and associated circulation pattern (e.g., a thermal low) and the analogue method could interpret this situation as circulation-induced rather than thermodynamic. Nudging all vertical levels of the atmosphere suppresses influence from the land surface to the atmosphere, so land-atmosphere interactions are likely to remain in the thermodynamic component of the piControl-nudged runs used as a benchmark in this study (Merrifield et al., 2019). |
| DEA | Because the approach removes the total effect of GMST without conditioning on land–atmosphere interactions, it may also eliminate the mediating effect of GMST operating through this pathway (but this effect is likely small and confined to trends that are colinear with GMST), whereas the mediating effect of atmospheric circulation is expected to be retained, given that the linear model has sufficient expressive capacity to capture these complex relationships. |
| UNET | The SLP is used as a predictor of the circulation. However, this variable may contain surface imprints which might affect the "circulation-induced" component. Therefore, land-atmosphere interactions may be partly predicted by the UNET architecture. |
| nudged simulations | Nudging is expected to separate the land-atmosphere interactions into a thermodynamically-driven, and a circulation-driven component (i.e., atmospheric imprints of land-atmosphere interactions are expected to be captured through nudging). |

Neither of these examples particularly undermines the nudged simulations but do highlight how they are fundamentally different from the statistical approaches, as they implicitly include more thermodynamic effects adn feedbacks.

This may explain why the distibutions of the trends are systematically underestimated (e.g. the distrubutions on the right of Figure 3 and in Figure D2) in all the statistical approaches as they

do not include the feedbacks and adjustments that are implicit in the nudged runs. At present there is only a discussion of the limitations of the statistical methods but I think discussing the limitation fo the nudged approach would also be useful to include.

*We added a discussion of the limitations of the nudged circulation experiments and changed the framing accordingly (see previous comments). Concerning the systematic underestimation of circulation induced trends in statistical decomposition methods, we are quite confident that they are mainly due to the underdispersiveness of the models. We cannot exclude that the mentioned shortcomings are relevant.*

*Line 369-374: "Concerning the land-atmosphere interactions, we conclude that the effect on our estimates of circulation induced trends is diverse between methods. This increases our confidence in the signals all methods agree on (e.g. circulation induced warming over Europe). At the same time, there is no systematic (and consistent) difference in how statistical decomposition methods might be affected by land-atmosphere interactions in comparison to how land-atmosphere interactions might affect the nudged simulations. Therefore, the effect of land-atmosphere interactions cannot explain the systematic underestimation of the magnitude of circulation induced trends in statistical decomposition methods (as compared to the nudged simulations)."*

I am sure the authors can directly discuss and address these differences and I think this would strengthen the interpretation of the results.

Minor comments:

Figure 3: This is a bit messy in the version I have– more details on the KDE plots on the right would be usefukle (along with axis labels etc).

*We agree that axis labels in the KDE plots are required. We did not find a solution to include in a size that is still readable and removed them from the plot. KDE plots are still shown in the appendix.*

Section 2.3.4: I don't quite follow why the SLP would not be detrended as the "forced response" is small. It this important? Surely it would be better to include, unless the results are sensitive to this? I also may have misunderstood this, in which case a brief clarification might help.

The objective is to estimate the part of daily temperature variations which can be explained by the large-scale circulation (using the SLP as a proxy of this circulation).

As we are working with historical+SSP runs, we need to account for the forced response in the temperature which is not insignificant. Thus, for the training, we remove from the temperatures the mean seasonal cycle and the estimation of this forced response (with the method described by Rigal et al, 2019) as they are not circulation induced. We did not pre-process the SLP data

by removing an estimate of the forced response because it is small in the SLP (Figures 2 and 3 show that the three piControl-nudged experiments do not exhibit significant common trends).

I want to end on a positive: I think this is a very interesting paper!

---

## Author Response (AR2)

**Response to editor comments**

The first referee is satisfied with the revisions. The second referee was not available to check the revisions, but I have assessed whether their concerns were addressed and believe they are. Overall, the authors have done a nice job of revising the manuscript to improve the clarity and interpretability of the results.

Below I've listed some issues requiring clarification and/or correction that seem to have slipped through, some of which are related to points from the ECR peer-review training group. In addition, I strongly recommend a dedicated proofreading pass to catch the small but numerous writing glitches throughout the manuscript. The goals/results of this study should be very interesting and useful for the community, hence I feel these final efforts are worthwhile. The manuscript should be ready for publication after these points have been addressed.

Thanks a lot for providing another review!

- The peer-review training group had a comment about the mismatch between the "main objective of the study ... to investigate long-term temperature trends" and the "the prominent motivation based on heatwaves and extremes" (i.e., the first two paragraphs of the intro). The revised manuscript now includes some lines in the second paragraph to bring the focus back to trends, which goes some way to addressing this problem. I think it could be fixed completely by doing a bit more reorganizing/refocusing of the first two paragraphs - perhaps lead with the trends and bring in the extreme heatwaves as an important related impact. (The peer-review group had a different suggestion, which would also work but would take things in a different direction: "It would be helpful if the authors could discuss a more concrete example of how their estimated trends allow a better understanding of extreme events").

We agree, that heavily focusing on heat extremes in the beginning of the introduction may be a bit misleading. We slightly changed the first paragraph and have included more text about warm seasons and heat waves. Overall, we believe that the focus on warm seasons (including heat waves) is still a good motivation for the trend analysis, because the heat waves and warm seasons are indeed an important impact of the trends (and were studied in that way also for instance in Teng et al., 2022).

- L101: The use of the word "experiments" here confused me. I believe you've run 3 simulations (or members) of the historical experiment and 3 simulations of the SSP370 experiment, correct? Sometimes, you refer to the historical and SSP370 simulations as different runs, and sometimes you refer to the combined hist+SSP370 simulations as one transient run. Please pick one and stick with this. Table 1 still refers to 1300, 1400, etc.

We replaced "experiments" by "simulations" in most instances to avoid confusion. We also adapted table 1. Thanks for noting the inconsistency!

We also introduce the scenario hist+SSP370 properly when it is first mentioned (line 73-75).

- Section 2.2: The description of these experiments is confusing. I
think you should state upfront that there is a piControl and a
hist+SSP370, and clarify what you mean by "the same" anthropogenic
forcing (L179). Are they all run in AMIP configuration with observed
SSTs?

We clarified the description of the experiments and the used forcing. We also clarified the AMIP
setup and highlighted the differences to the other nudged experiments.

- Section 2.3: The peer-review training group had some very nice
suggestions here, and I feel they should be straightforward to
implement (the revised manuscript already reflects some of these
suggestions). Readers who do not want to go into the details should be
able to read the start of each subsection and still appreciate the
study's results. In addition, a few comments from me on the revised
version:
* 2.3.1 the info in L209-213 is useful for introducing the method,
\lambda is called both the ridge regression parameter and the
regularization parameter).

Thanks for the suggestion. Changed.

* 2.3.2 has improved in response to referee 1's comments, although I
don't understand L251-252, and the last paragraph remains quite
confusing.

The paragraph has been re-written as:

In this study, analogues are selected from the free-running hist+SSP370 simulations to dynamically
adjust (1) each hist+SSP370 simulation and (2) ERA5. Each hist+SSP370 simulation is dynamically
adjusted using the ``leave-one-out'' approach (Deser et al., 2016; Lehner et al., 2017). In the leave-
one-out approach, for each month, e.g., June 1900, analogues are selected from all other Junes in
the simulation's 1850-2014 period except 1900. The leave-one-out approach is used for the
comparison between the hist+SSP370 and piControl-nudged simulations. In the second approach,
analogues are selected from the entire 1850-2014 period of each of the hist+SSP370 simulations
and used to dynamically adjust ERA5. The resulting three dynamical components of ERA5 are
shown in Figure F1 and are averaged to produce the circulation-induced trend estimates in Figure 4.

* 2.3.3 similar to 2.3.1, could use a general description before
launching into details.

We added short introductory descriptions for all methods.

* 2.3.4 What are the "8" CESM2 transient
simulations? We've only heard of 3 x 2 experimemts = 6 until now?

In order to avoid over-fitting, the UNET is trained on 8 other transient simulations (historical + ssp) from the CESM2 ensemble, i.e. different from the 3 simulations that have been used to build the nudging experiment (members 1300, 1400 and 1500). This has been clarified in the revised version.

- L343-354: I appreciate the additional details here, but it would be clearer if you made these an enumerated list so the reader gets the explanation of each metric right away.

Good idea. We changed it to an enumerated list.

For (i), what is meant by "the fraction"? Is this just spatial, i.e., out of total gridpoints? This should also be clarified in the Table 1 caption. * ah yes, explained in L372 *

We clarified it in the description at the beginning of the results section and it the caption of table 1.

For (ii), is it an area-weighted spatial correlation?

No, nothing is area weighted here. We added a comment on this in line 273.

Other (language, technical, etc.)

- Check peer-review group's various suggestions.

Done

- L16: Suggest "sign" rather than "direction".

Done

- L94-97: I like the addition of some preamble to the Data & Methods section, but this makes it sound like the decomposition methods are only applied to ERA5!

We added a sentence clarifying that the decomposition methods are evaluated on CESM2 simulations.

- L107: Is it just the forcing that follows the CESM2 LENS2 protocol, or also the method of generating initial conditions (for example, do the 3 members include different ocean states, as in LENS2, or just atmospheric perturbations)?

The three initial conditions used for the freely running hist+SSP370 (and their corresponding nudged simulations) are from a piControl run and the year of initialization for each run is separated by 100 years: namely year 1300, year 1400 and year 1500. Therefore, they have different ocean states. We added a comment concerning the initial conditions in the manuscript (line 76-78).

- Figure 2: Can you please explain the various lines in panel a in the caption? The figure label says 50-year smoothed GMST, which I assume are the bold lines. The thin lines are annual means?

Changed the figure label and caption.

- L155: "both of these processes" suggests only 2 processes at play - suggestion "both thermodynamic and dynamic contributions"

Changed

- L172: It's a bit odd to mention ridge regression and DEA results here, before the methods have been described. I see why this point was added here, but it can be done more generally (referencing some of the decomposition methods, for example)

Changed

- L405-6: Last sentence, second part is quite vague and makes the entire sentence a bit confusing.

Changed

- Some broken labels throughout (figures, references)

- I would tend to favour "smaller/weaker vs larger/stronger" trends, as opposed to "lower vs higher" trends.

Agreed and changed

**Response to:**

Review by Arundhati Kalyan, Anjali Thomas, and Jan Zibell*

*J.Z. declares being employed at the same institute as one of the co-authors of the study.

We copied the review and added our responses in **green**.

In the above study, the authors assess long-term (multidecadal) circulation-induced changes in summer temperature trends in the northern hemisphere midlatitudes (30–60°N). They employ four different statistical/machine learning-based methods – ridge regression, atmospheric circulation analogues, direct effect analysis, and a convolutional neural network – to isolate temperature trends driven by circulation changes over historical and future time periods in free-running climate model simulations (using CESM2) and ERA5 reanalysis. The relative performance of these methods is evaluated against a benchmark comprising nudged climate model experiments (also using CESM2) which include forced and internal components of circulation variability, but no forced thermodynamic component. The four methods are found to be effective on climate timescales, albeit with biases. The paper also highlights regional differences in dynamically forced trends, revealing alternating wavelike patterns of warming and cooling throughout North America and Eurasia. Finally, the authors discuss in depth the challenges and limitations in using statistical methods to decompose circulation vs. thermodynamically driven trend signals.

The study makes robust choices relating to data and methodology, in that four different decomposition methods are used and multiple statistical evaluation metrics are analysed for each. The authors transparently present a summary of the climate of the wind-nudged simulations which form the reference for their assessment of the decomposition methods. The presentation of challenges (multidecadal timescales, differing climate components included in the circulation-related decomposition, creation of a suitable benchmark, need for multiple models) in the discussion section is strong and may serve as useful reference for future studies. The study completes the stated target of quantifying circulation-driven trends across the NH midlatitudes and validating the different methods against a suitable climate model benchmark. In addition to highlighting features with low and high skill for each method, the study adds some degree of confidence to the findings of earlier studies that examined regional patterns of circulation-driven temperature trends. The article is concise and generally well-written. The title is informative and appropriate to the study, although the abstract could be improved as suggested below. The relevance of the study,

objectives, and scientific challenges are introduced well. Even so, there are portions of the manuscript that are hard to follow and hinder conceptual understanding and interpretation of the results. In particular, the explanations of the statistical methods, the illustration of statistical significance/uncertainties associated with the results, and the dynamical interpretation of the study could be improved. We recommend publication of this manuscript in Weather and Climate Dynamics after minor but necessary revisions as outlined in the comments below.

We thank the reviewers for taking the time to thoroughly read and comment our manuscript. The raised concerns and suggestions are very valuable and helpful for the finalization of the paper.

We received these reviews after completing a revised manuscript addressing comments from two other reviewers. Therefore, some of the comments are already discussed in the response to reviewer 1 and reviewer 2. Furthermore, some suggestions have already been implemented in the process of preparing the revised manuscript. We mainly answered comments where we aim to clarify aspects. We also briefly commented on suggestions that we implemented in the final version. In few cases we do not reply, because in these cases the aspects have been largely addressed already at the revision stage. Please reach out to us if you are interested in more detailed answers to specific comments.

Statistical methods: The description of the four methods used is not straightforward and quite hard to follow for a non-expert reader. We are not experts in the statistical/ML methods used in this study and hence, cannot comment on the strengths and weaknesses of the design and implementation of these methods. However, here are a few suggestions to make the methods understandable to a broader dynamics audience, such that we can better appreciate the significance of the study's findings:

- Add an introductory sentence or two in plain language in all of the sections 2.3.1–2.3.4 to explain what the method and/or equation aims to achieve

  Section 2.3.2: Great recommendation! We've added the sentence: "It achieves this through the re-construction of monthly mean climate fields using linear regression with coefficients derived from a field representative of atmospheric circulation (here, sea-level pressure)."

- Clearly define all variables and constants introduced in each equation

  We believe that all variables and constants are now introduced.Consider also whether certain details in the method descriptions can be moved to supplementary text

- Here a few line-specific comments:

  - Line 134: Please explain why a 40°×40° rectangle around each grid cell was chosen. Could other sizes change the results?

    This is a good question and usually a sensitivity analysis would be useful. In the case of the ridge regression, the coefficients of grid-cells far away from the location of interest are kept small by the regularization and therefore, the results are not expected to be sensitive to the extent of the region.

  - Line 156: Please discuss briefly how you choose the number of analogues ($N_s = 50$) and repetitions ($N_r = 100$). Does changing these choices affect the results?

    Great question, we have added that these numbers were selected to be consistent to how the method was in previous studies. If you are interested, a more comprehensive sensitivity analysis for another domain is available here: https://escholarship.org/uc/item/5mz52654

  - Line 200: It might help to briefly explain the physical meaning of the $Y_{perp}$ component in plain language in this context.

It would be good to see more rigorous discussion of the statistical significance and uncertainties associated with the results presented. The need for this arises due to the following reasons:

The question about the significance of the trends has also been raised by reviewer 1. Please see our response to reviewer 1, which we hope addresses your concerns raised above and regarding the specifics below.

- "… observed trends are falling out of the range of model-simulated expected trends" (Line 35)

- There is "ambiguity" on the magnitude of the historical circulation-induced trends, and all methods likely underestimate these magnitudes (Sec. 3.2)

- In light of the above, it would be helpful to better understand the likely range/distribution of trends for each method and how they differ. Are some methods more likely to include the observed/nudged values than others?

- The use of a very small sample size (only three ensemble members) for the free-running and nudged simulations limits the ability to show robust

uncertainties. Could you provide more information on whether the initial conditions for the three nudged ensemble members were chosen at random or based on certain criteria? Could you comment on how large of an added value more members would be and why you decided on three only?

We use of the nudged simulations as test cases that are supposed to mimic the intended application of the decomposition methods. The decomposition methods are usually applied to reanalysis data. With our test cases, we stick to the application to one climate trajectory as it would be done in reanalysis. We will not be able to give meaningful confidence intervals for the estimated trends in ERA5 but we want to know how meaningful the estimated trend pattern is (and magnitude of the pattern). For this purpose, three members are sufficient. The initial conditions were chosen to be mostly independent in terms of ocean states. Lastly, we would like to comment on the ensemble size in general: We agree with the reviewers that in principle, more members would be better (of course). However, with our specific setup, we believe that three members are indeed enough to sample the thermodynamical component well, because the thermodynamical component for each of the three simulations falls very close to the ensemble mean. More importantly, the dynamical component can be used as a test case for the nudging, and for this application we strongly believe that the three members are sufficient. Yet, of course we agree that more members would be useful for instance if the goal would be to identify potential forced dynamical responses.

- Table 1 would also benefit from the inclusion of confidence intervals and/or p-values to reject the null hypothesis that the correct sign or correlation of temperature trends was obtained by random chance.

This point was also raised by reviewer 1. Please find a response to reviewer 1 comment 2.

There are instances where we think that the discussions in this study may benefit from a more dynamical perspective:

- It is clear from the introduction and methods that the wind-nudged simulations are viewed as a ground-truth benchmark. The limitations of this approach are discussed as, for instance, in line 335: "However, there may be factors of residual climate variability (such as ocean variability) or feedbacks between circulation and other factors such as land-atmosphere coupling that could still affect

thermodynamical processes on climate over land." Aren't there even more concrete examples for a physical relationship that is not a priori captured, such as the time mean thermal wind balance?

This was also the main concern of reviewer 2. Please check the response to reviewer 2.

- Line 36–37: Please consider to slightly reframe "... may indicate ... that a forced change in circulation is missing in the models". This framing could make one think of unresolved/parameterized processes in climate models, such as latent heating in deep convection in the midlatitudes. For those it is the consensus that a forced circulation change *is* missing in the models. Should you actually refer to this, the sentence could be reframed as "that circulation changes due to unresolved processes, which are not captured by the models, turn out to be meaningful / non-negligible".

Since in this study we do not study the reasons for misrepresented processes in climate models, we actually prefer to keep the statement vague and general.

- It is certainly not the purpose of this study to discuss all limitations of wind-nudging in detail and of course every alternative also has its limitations, but we suggest that the authors at least address the point that in the real atmosphere, the development of a heatwave is not the linear sum of a thermodynamic component plus a dynamic component. There are many non-linear dynamical feedbacks from thermodynamical processes, e.g. from coupling with radiation via clouds, surface fluxes depending on soil moisture, or latent heat release (which, if speaking of heat waves, within a warm conveyor belt may increase blocking intensity (Pfahl et al. 2015)). Possibly, addressing this is related to emphasizing more prominently that the authors regard GMST as the representative variable of thermodynamic changes.

The reviewers raise an important point that was also raised by reviewer 2. In the revised manuscript we discuss the implications of the highly simplified decomposition into "circulation-induced" and "thermodynamic" in more detail.

- Overall, we suggest that 1) the assumption that a trend can be decomposed into thermodynamical and dynamical components and 2) the use of wind-nudged simulations (as introduced in Sect. 2.1) to achieve this are discussed a bit more critically. This could be done very briefly when introducing the wind-nudging experiments in the Introduction and then in a more elaborate way for

instance as a sixth discussion point in Sect. 3.3.

Thanks for this very good suggestion! This is what we ended up doing in the revised manuscript.

The main objective of the study is to investigate long-term temperature trends. Therefore, the prominent motivation based on heatwaves and extremes seems somewhat out of place. Motivating this research with individual events is fine per se, but this study does not investigate trends in heatwaves. It would be helpful if the authors could discuss a more concrete example of how their estimated trends allow a better understanding of extreme events (as indicated in lines 31-34)?

Thanks for this comment. We updated the beginning of the introduction by shifting the focus more towards warm seasons and their impacts.

The use of the dynamical vs. thermodynamical separation of trends could be even further strengthened by a discussion of the geographical variability of the thermodynamically induced trends. In Figure 4, over Eurasia your methods disagree whether the thermodynamical change is rather uniform or dependent on longitude or latitude. Are there any physical arguments in the literature for what the thermodynamically-induced pattern of warming should look like? For instance, it is observed that the Mediterranean region is warming faster (Brogli et al., 2019). Alternatively, it seems also fine to note that this is left for future study or refer to discussions in other studies.

This is a very interesting observation and we will add a comment about this difference in the estimates for the thermodynamic contribution in the discussion where we discuss the differences in how decomposition methods separate between "dynamical" and "thermodynamical" (line 367-368).

**Minor comments:**

•Line 3–4: "Over the northern hemispheric mid-latitudes, considerable regional differences in summer temperatures have been observed." → Presumably, you mean differences in summer temperature *changes.*

Has changed.

•Line 5–6: We think the general readability of the abstract would benefit from a brief description of 'decomposition method', i.e., what you decompose the trends into. If one is not very familiar with the topic or similar literature, this is not obvious but only (and well) presented in the introduction.

Good idea, done.

•Line 10–11: "Most decomposition methods show skill in estimating the sign of circulation-induced trends but all methods underestimate the magnitude of these trends." This statement contains the fact that you use the wind-nudged simulations as your benchmark and that you assume that the nudged simulations contain 100% of the dynamical component of the trend. This should be presented more clearly as was done in the Introduction, for instance, around line 60.

As discussed later in the paper, we are convinced that this underestimation of the magnitude of trend patterns is not affected by limitations in the nudging experiments as benchmark.

•Line 16: Consider changing: The intensity of heatwaves "increases globally" to "has been increasing globally".

•Line 18–19: Consider adding a reference/s here to show that intensification of heat waves occurs in a warmer climate due to thermodynamic factors (perhaps an attribution study?).

•Line 19: "However, heat waves are not only…" is a long sentence and could benefit from restructuring.

Changed

•Line 22: It might be worth mentioning here whether land-atmosphere interactions are more or less important than circulation changes as a factor in driving summer temperature trends.

•Line 23: It might be worth commenting on whether regional trends are more pronounced in the NH midlatitudes than elsewhere. May also be good to include a sentence citing studies that examined trends in the tropics or Southern Hemisphere.

•Line 24–27: This is a long sentence and could be split into two separate sentences.

•Line 31: Consider including more specific references that show why forced changes in circulation are small compared to internal variability.

•Line 48: Consider whether there are any other limitations of the nudged circulation experiments and include that here.

•Line 49: Add "e.g. the circulation not being in thermal wind balance" to clarify that you don't mean unresolved processes, which are also mechanisms not represented in the models used.

We added a note on the limitations of nudging and refer to the discussion where we add an example of potential inconsistencies in line 378-380.

•Line 49–51: On the other hand, most of statistical decomposition methods …" can be rewritten/shortened or split into two sentences to enhance readability.

Changed

•Line 54–56: "Moreover, benchmarks for circulation-induced long-term trends have not been available so far, and to our knowledge no systematic comparison of dynamical adjustment methods has been performed." It would be good to clarify if this applies globally or just to mid-latitudes and to briefly mention if any emerging efforts exist. This would make it clearer why the study is filling an important gap.

This statement is quite general because – to our knowledge – such benchmarks haven't been used so far in any region.

•Line 57: Add the specific NH latitude range being examined here.

Done

•Section 2: Adding a sentence or two linking each data/methods subsection to the main aim (separating thermodynamic vs circulation-driven temperature trends) would help the reader understand why each method or simulation is being used.

•Line 69: You could specify the ERA5 years here as well.

Done

•Line 72: "First, three standard historical and future forcing experiments …." At first reading, this sounds like three different forcing scenarios or the like. Please specify that you mean three ensemble members.

We changed the wording to "simulations".

•Line 80: How about specifying some of the main features of the nudging, e.g., whether your nudging is done at the model grid or involves some spectral transformations, and to which vertical level it is done? This way the reader gets a good first impression without having to refer to Topal and Ding (2023) to find out what is "similar" to their approach and what is different.

Thanks, we have added that the nudging involves regular relaxation procedure applied onto the horizontal winds (and all the levels are described). We now just say that the nudging procedure was used in previous studies such as Topal and Ding (2023).

•Line 80: Is CAM6 an abbreviation of something? If yes, please mention.

Thanks, implemented (Community Atmosphere Model Version 6)

•Line 81: "These simulations will be henceforth referred…". Simplify this sentence for readability.

Done

•Line 87: The phrasing of thermodynamic forcing being represented by "surface temperature" somewhat suggests that using this metric of forcing is not a choice. In reality, temperature change is non-uniform throughout the atmosphere with implications for the midlatitude circulation. We suggest that the authors instead say 'commonly approximated by GMST' or similar.

Done

•Line 89: Good point, but it would help to add a short explanation of why it is hard to evaluate these methods in a coupled system.

We discuss this issue in depth in the revised manuscript (see response to other reviewers' comments on this aspect)

•Line 95: Will the residual internal variability (e.g. from the ocean) influence the evaluation of the decomposition methods, and how do you account for that?

Is discussed in the following paragraph.

•Line 111: "However, we assume that the effect…": This assumption is reasonable, but you might want to add a reference or a short justification for this assumption.

•Figure 2: It is not clear what is meant with the cooler versus warmer histograms in panels b), c). Please clarify.

•Line 115: Consider splitting the paragraph into two shorter ones: one describing the experimental setup and another explaining its implications and limitations. This would improve readability.

•Line 120: Briefly define AMIP in the text for clarity.

Done

•Line 132: Could split into two shorter sentences for clarity.

•Line 216: Consider providing more details of the transient CESM2 simulations used to train the UNET.

Done

•Lines 219–220: Clarify why the training is done on CESM2 first and then fine-tuning on ERA5—why does this improve performance or robustness?

Done

•Line 220–222: Rephrase to sound more concise and formal.

•Line 227–237: Consider presenting these two sets of bullet points together instead of separately to make the text more concise and easier for the reader to associate each skill metric with what it represents.
Good idea, done.

•Section 3.1: The discussion mentions how the methods differ (DEA captures magnitude, UNET conservative), but the rationale behind these differences could be explained more clearly. For instance, why does UNET underestimate magnitudes?

This is a typical pattern in statistical or machine learning methods that are trained with a loss function that minimizes mean squared error (= the bulk of the distribution, which pays a price at the tails).

•Line 276: This is a long sentence. Consider splitting it into 2–3 smaller sentences to enhance readability.

•Line 289–291: In "The ridge regression … up to 0.6 K/dec", change "where" to "were", and add "*suggest* stronger circulation induced trends …"
•Fig. 4: Which wavelength could approximate the wave-pattern change that you find? Can you relate this to other studies?

•Line 305: This paragraph sounds like a re-introduction of dynamical adjustment from zero. A bit of repetition is appreciated for the flow, but at the current stage this introduction of dynamical adjustment is even clearer than in the introduction (using even more references). Please consider streamlining this or, otherwise, stating more clearly if you mean something different than in the introduction or possibly moving some of this material into the introduction.

•Section 3.3 is purely a discussion. Why not make it a new section called Discussion? There is no new result in this section.

Good idea, changed.

**Technical comments:**

•Line 35–36 and onward: Check for the use of citet vs. citep and citep[][]{} throughout the paper.

•In Figure 3, the kernel density maps could be enlarged with axes labels shown.

Removed

•Figure A2 is not explained or referenced in the text. Change.

Removed

•Figure B2 is not explained or referenced in the text. Change.

Checked

References:

Brogli, R., N. Kröner, S. L. Sørland, D. Lüthi, and C. Schär, 2019: The Role of Hadley Circulation and Lapse-Rate Changes for the Future European Summer Climate. J. Climate, 32, 385–404, https://doi.org/10.1175/JCLI-D-18-0431.1.

Pfahl, S., Schwierz, C., Croci-Maspoli, M. et al. Importance of latent heat release in ascending air streams for atmospheric blocking. Nature Geosci 8, 610–614 (2015).

https://doi.org/10.1038/ngeo2487